# TrainRef: Curating Data with Label Distribution and Minimal Reference for Accurate Prediction and Reliable Confidence

**Murong Ma[1],*, Ruofan Liu[1],†, Yun Lin[2],‡, Zhiyong Huang[1], Jin Song Dong[1]**
[1]School of Computing, National University of Singapore
[2]School of Computer Science, Shanghai Jiao Tong University

## Abstract

Practical classification requires both high predictive accuracy and reliable confidence for human-AI collaboration. Given that a high-quality dataset is expensive and sometimes impossible, learning with noisy labels (LNL) is of great importance. The state-of-the-art works propose many denoising approaches by categorically correcting the label noise, i.e., change a label from one class to another, which can suffer from *normality pollution* and *class ambiguity*. The normality pollution happens when the noise ratio gets higher, leading to prediction inaccuracy, as such approaches intrinsically learns normality from the noisy dataset. The class ambiguity happens when the number of classes increases, leading to less reliable prediction confidence.

In this work, we propose a training-time data-curation framework, TrainRef, to uniformly address prediction accuracy and confidence calibration by (1) an extrinsic small set of reference samples $\mathcal{D}_{\text{ref}}$ to avoid normality pollution and (2) curate labels into a class distribution instead of a categorical class to handle sample ambiguity. Our insights lie in that the extrinsic information allows us to select more precise clean samples even when $|\mathcal{D}_{\text{ref}}|$ equals to the number of classes (i.e., one sample per class). Technically, we design (1) a reference augmentation technique to select clean samples from the dataset based on $\mathcal{D}_{\text{ref}}$; and (2) a model-dataset co-evolving technique for a near-perfect embedding space, which is used to vote on the class-distribution for the label of a noisy sample. Extensive experiments on CIFAR-100, Animal10N, and WebVision demonstrate that TrainRef outperform the state-of-the-art denoising techniques (DISC, L2B, and DivideMix) and model calibration techniques (label smoothing, Mixup, and temperature scaling). Furthermore, our user study shows that the model confidence trained by TrainRef well aligns with human intuition. More demonstration, proof, and experimental details are available at `https://sites.google.com/view/train-ref`.

## 1 Introduction

Practical classification application, such as medical diagnosis (Rosenbacke et al., 2024), autonomous driving (Delavari et al., 2025), and fraud detection (Perini & Davis, 2023), requires both accuracy and reliable confidence. Recent work by Kalai et al. (Kalai et al., 2025) shows that even highly capable LLMs tend to produce overconfident false predictions (hallucinations), emphasizing the importance of calibration as a peer to accuracy. The confidence is useful for the model users to decide when to adopt model decision (Corbière et al., 2019; Pan et al., 2020). Such model performance (both accuracy and confidence) usually requires high-quality datasets, however, which are usually expensive, sometimes impossible (AI, 2024; Forbes, 2024).

Therefore, learning with noisy labels (LNL) solutions emerge to address the challenge. The solutions evolve from label transition matrix (Hendrycks et al., 2018; Patrini et al., 2017), sample-reweighting techniques (Li et al., 2020; Sheng et al., 2024; Liu et al., 2020), gravitating to the pseudo-labeling

---

*Part of this work was conducted during a visit to SJTU.
†Corresponding authors: `lin_yun@sjtu.edu.cn`, `liu.ruofan16@u.nus.edu`

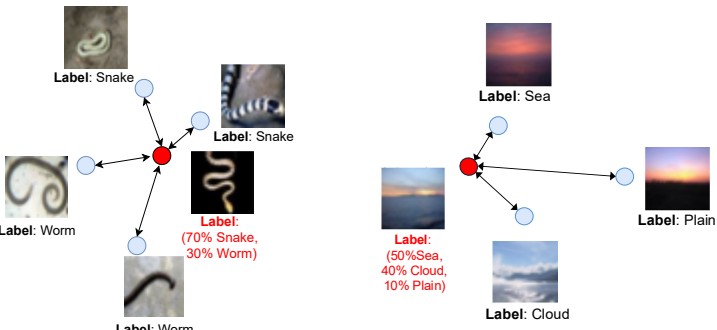

Figure 1: Examples of ambiguous samples from CIFAR-100 where the ground-truth label is distributional rather than categorical. (Left) Snake vs. Worm, where the label can be 70% Snake and 30% Worm. (Right) Sea vs. Cloud vs. Plain, where the label may be distributed as 50% Sea, 40% Cloud, and 10% Plain. Categorical labels for such cases encourage overconfident predictions, whereas distributional labels better capture inherent ambiguity.

techniques which curate the noisy data from one class to another via semi-supervised learning (Kim et al., 2021; Nishi et al., 2021; Lu et al., 2021; Zheltonozhskii et al., 2022; Li et al., 2023). While generally effective in improving predictive accuracy, the performance of such categorical curation (i.e., change one label with another) to learn reliable confidence is limited, especially when the number of classes grows, leading to more ambiguous samples (see Figure 1). In practice, to classify a clinical note into an ICD code, there can be more than 10K classes. In addition, those solutions curate the labels by learning the label normalities from the *intrinsic* noisy dataset. When the noise ratio is high enough, the noisy samples can form the *polluting* normalities, leading to mis-curation. On the other hand, confidence calibration solutions include label smoothing (Müller et al., 2019), mixup (Zhang, 2017), and temperature scaling (Hinton, 2015; Guo et al., 2017) which mitigates prediction overconfidence for more reliable class distributions. However, their performance is limited in addressing the mis-confidence caused by label noise.

In this work, we propose a training-time data-curation solution, TrainRef, with the following features.

- **Distributional Curation**: Different from categorical curation, we learn distributional curation during training, i.e., change a label from one class to a class distribution, for having high predictive accuracy and reliable confidence in a uniform manner.

- **Extrinsic Reference Set**: Different from the state-of-the-arts (Li et al., 2020; Sheng et al., 2024; Liu et al., 2020; Li et al., 2023) which curate the labels by learning the label normalities from the *intrinsic* noisy dataset, our approach introduces a tiny trusted set of reference samples with the ground-truth label, $\mathcal{D}_{ref}$, as label normalities, to avoid normality pollution caused by high noise ratios. Our approach can be effective even when $|\mathcal{D}_{ref}|$ is as small as the number of classes (one sample per class).

Given a trusted reference set $\mathcal{D}_{ref}$ and a noisy dataset $\tilde{\mathcal{D}}$, TrainRef adopts a three-stage training routine by co-evolving the model embedding space and the curated dataset. First, we obtaining a label-agnostic encoder $\theta$ by pre-training masked-image modelling (MIM) (Pathak et al., 2016; Peng et al., 2022) task on $\tilde{\mathcal{D}}$. As a result, the embedding space of $\theta$ is immune to noise. Next, we design a reference augmentation technique to select clean samples $\hat{\mathcal{D}}(\hat{\mathcal{D}} \subset \tilde{\mathcal{D}})$ based on $\mathcal{D}_{ref}$, regarding the agreement between the samples in $\mathcal{D}_{ref}$ and the samples in $\tilde{\mathcal{D}}$ through the influence functions (Koh & Liang, 2017; Pruthi et al., 2020; **?**) computed on $\theta$. Finally, we co-evolve the model $\theta^*$ and the dataset $\hat{\mathcal{D}}$ by iteratively (1) fine-tuning a model $\theta^*$ with learned clean dataset $\hat{\mathcal{D}}$ and (2) curating and enhancing the dataset $\hat{\mathcal{D}}$ by voting the label distribution for noisy samples with their neighbouring clean samples on the embedding space of $\theta^*$.

Our extensive experiments show that TrainRef significantly outperform the state-of-the-art denoising solutions (i.e., L2B (Zhou et al., 2024), DISC (Li et al., 2023), LSL (Kim et al., 2024)) by 1.82% to 8.19% across various benchmarks , and the state-of-the-art confidence calibration solutions (i.e., label smoothing (Müller et al., 2019), mixup (Zhang, 2017), and temperature scaling (Hinton, 2015;

Guo et al., 2017)) by consistently achieving lower ECE indicating better calibration. Furthermore, in a blind user study, participants agree with TrainRef's confidence estimates over previous SOTA (DISC + mixup) in **75%** of cases, corroborating its reliability in practice. More demonstration, proof, and experimental details are available at `https://sites.google.com/view/train-ref`.

## 2 PROBLEM STATEMENT

Given a collected dataset $\tilde{\mathcal{D}} = \{(\mathbf{x}, \tilde{\mathbf{y}})\}_{i=1}^{N}$ where each sample consists of an input $\mathbf{x} \in \mathbb{R}^d$ and a one-hot categorical label $\tilde{\mathbf{y}} = [y_1, y_2, ..., y_C]$, $y_c \in \{0, 1\}$, $C$ is the number of classes. We assume that the true label is a class distribution instead of a one-hot class, that is, $\mathcal{D}^* = \{(\mathbf{x}, \mathbf{y}^*)\}$, where $\mathbf{y}^* = [y_1^*, y_2^*, ..., y_C^*]$, $y_c \in [0, 1]$, $\sum_{i=1}^{C} y_c^* = 1$. Thus, we define two types of label misinformation:

- **Categorical Noise**: The ground-truth label $\mathbf{y}^*$ is one-hot (e.g., a 0-entropy distributional label) but $\arg\max_c \mathbf{y}_c^* \neq \arg\max_c \tilde{\mathbf{y}}_c$.
- **Distributional Noise**: The ground-truth label $\mathbf{y}^*$ is a Dirichlet distribution over $C$ classes, and $\mathbf{y}^* \neq \tilde{\mathbf{y}}$. Intuitively, such noise can lead to an over- or under-confident model.

Our goal is to learn a parameterized mapping $f_\theta(.) : \mathbb{R}^d \to \mathbb{R}^h$ that projects each input into an embedding space for downstream tasks. We obtain our estimator $\hat{\theta}$ by minimizing the empirical risk over a finite dataset with distributional labels (Equation 1). Here, $\lambda \|f\|$ denotes a regularization term, such as weight decay, that penalizes large parameter values to control the model's capacity.

$$\hat{\theta} = \arg\min_\theta \frac{1}{N} \sum_{(\mathbf{x}_i, \tilde{\mathbf{y}}_i) \in \tilde{\mathcal{D}}} \left[ \mathcal{L}\big(f_\theta(\mathbf{x}_i),\ \tilde{\mathbf{y}}_i\big) + \lambda \|f_\theta\| \right] \tag{1}$$

In the limit as $N \to \infty$, a well-behaved estimator $\hat{\theta}$ should converge to the true risk minimizer.

$$\theta^* = \arg\min_\theta \mathbb{E}_{(\mathbf{x}_i, \mathbf{y}_i^*) \sim \mathcal{D}^*} \left[ \mathcal{L}\big(f_\theta(\mathbf{x}_i),\ \mathbf{y}_i^*\big) \right] + \lambda \|f_\theta\| \tag{2}$$

However, in practice, this asymptotic consistency can break down when the dataset $\tilde{\mathcal{D}}$ contains label misinformation (i.e., both categorical and distributional noise).

**Rationale**  If the optimal embedding $f_{\theta^*}(.)$ were available, and $\hat{y} = g \circ f$, where $g(.) : \mathbb{R}^h \to \mathbb{R}^C$ is the classification head. Then by Representer Point Theorem (Schölkopf et al., 2001), any model prediction at query sample $\mathbf{x}_t$ can be expressed as a linear combination of representative samples $\mathbf{x}_i$, weighted by their similarity $k(\mathbf{x}_t, \mathbf{x}_i)$ (Equation 3), where each coefficient $\alpha_i$ depends only on the representative sample $(\mathbf{x}_i, \tilde{\mathbf{y}}_i)$.

$$\hat{\mathbf{y}}(\mathbf{x}_t) = \sum_{(\mathbf{x}_i, \mathbf{y}_i) \in \mathcal{D}_{ref}} \alpha_i \cdot k(\mathbf{x}_i, \mathbf{x}_t) \tag{3}$$

In TrainRef, (i) we estimate the optimal embedding function $\hat{f}$, and (ii) we collect a clean reference set with diverse class prototypes, $\mathcal{D}_{ref} = \{(\mathbf{x}_{ref}, \mathbf{y}_{ref})\}$. Then given a sample $(\mathbf{x}, \tilde{\mathbf{y}})$, we can curate the label as Equation 4.

$$\hat{\mathbf{y}}^*(\mathbf{x}_t) = \frac{1}{|\mathcal{D}_{\text{ref}}|} \sum_{(\mathbf{x}_{\text{ref}}, \mathbf{y}_{\text{ref}}) \in \mathcal{D}_{\text{ref}}} \mathbf{y}_{\text{ref}} \cdot k\big(\hat{f}(\mathbf{x}_{\text{ref}}), \hat{f}(\mathbf{x}_t)\big) \tag{4}$$

Here, we set $\alpha_i = \frac{\mathbf{y}_{ref}}{\|\mathcal{D}_{ref}\|}$, i.e. each clean label directly votes in proportion to its similarity with the query. We construct the curated dataset $\hat{D} = \{(\mathbf{x}_t, \hat{\mathbf{y}}^*(\mathbf{x}_t)) | \mathbf{x}_t \in \tilde{D}\}$ to replace $\tilde{\mathcal{D}}$ in the follow-up training processes, and we hope the $\hat{\theta}$ learnt from $\hat{D}$ is closer to $\theta^*$.

**Practical Challenges.** Applying Equation 4 in practice requires overcoming two challenges:

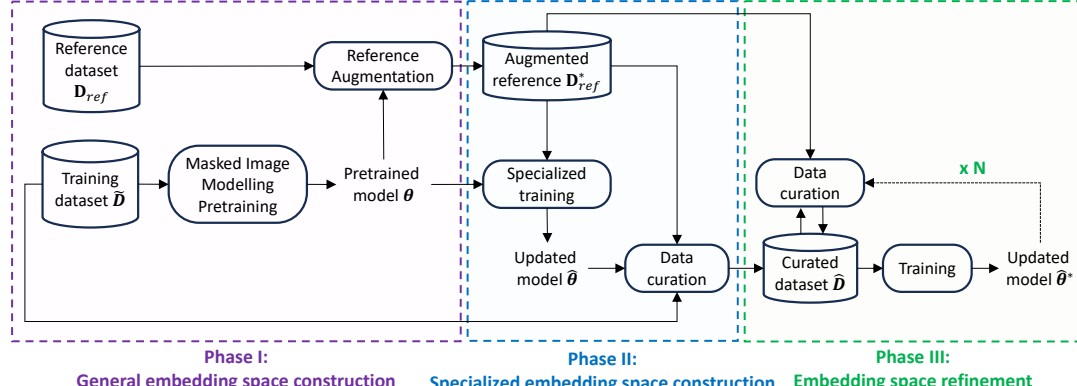

Figure 2: Overview of TrainRef: given a noisy training dataset $\tilde{\mathcal{D}}$ and a small reference dataset $\mathcal{D}_{ref}$, we (1) explore a near-perfect embedding space and (2) augment the reference to learn a function parameterized by $\hat{\theta}^*$ to minimize the empirical risk. The learned embedding space and the curated dataset are mutually influenced to converge.

- **Accurate embedding-space estimation.** Even with heuristics like the memorization effect (Liu et al., 2020) or confidence-based selection (Li et al., 2020; 2023), noisy or over-confident labels in $\tilde{\mathcal{D}}$ can skew the learned embedding space when minimizing the empirical risk (Eq. 1).

- **Reference-set diversity.** Manually verified clean references are typically few. We therefore need methods for augmenting and measuring diversity so that our reference pool contains sufficiently representative prototypes for reliable label voting.

Next, we introduce our solution TrainRef to address the above challenges.

## 3 APPROACH

Figure 2 illustrates the overall framework of TrainRef, which learns a deep classifier from a dataset $\tilde{\mathcal{D}}$ with both categorical and distributional noise, aided by a small clean reference set $\mathcal{D}_{\text{ref}}$. The core idea is to construct a reliable embedding space $\mathcal{S}^*$ and an augmented reference set $\mathcal{D}_{\text{ref}}^*$, such that samples in $\mathcal{D}_{\text{ref}}^*$ can collectively provide accurate supervision for the rest of the dataset via neighborhood voting. The training procedure consists of three iterative phases:

**Phase I: General Embedding Space Construction.** In this stage, we pre-train model $f_\theta$ from the training dataset $\tilde{\mathcal{D}}$ with loss functions independent from the labels. Specifically, we choose MIM (Masked Image Modelling) (Pathak et al., 2016; Peng et al., 2022) as the pretraining task which aims to recover a patch in an image sample from its neighboring patches. By this means, we learn an initial embedding space, allowing us to extract *general* semantically meaningful features from the samples.

**Phase II: Special Embedding Space Construction.** In the next stage, we augment the small reference set $\mathcal{D}_{ref}$ by introducing a subset of *clean* samples in $\tilde{\mathcal{D}}$. We consider a sample $(\mathbf{x}, \mathbf{y})$ in $\tilde{\mathcal{D}}$ as clean if $(\mathbf{x}, \mathbf{y})$ is consistent with all the samples in $\mathcal{D}_{ref}$. In this work, we introduce influence function (Koh & Liang, 2017) to measure such consistency (see Section 3.1). As a result, we have an augmented reference set $\mathcal{D}_{ref}^*$ which can be used to further retrain $\theta$ to $\hat{\theta}$. Comparing to $\theta$, the embedding space of $\hat{\theta}$ is more classification-relevant.

**Phase III: Embedding Space Refinement.** In the final stage, we iteratively update the learned model $\hat{\theta}^*$ and its embedding space and the curated model $\hat{\mathcal{D}}$. On one hand, we curate the dataset $\tilde{\mathcal{D}}$ to $\hat{\mathcal{D}}$ based on the learned embedding space and a set of discriminated reference samples. On the other hand, the curated dataset $\hat{\mathcal{D}}$ is further used to update the model to $\hat{\theta}^*$. The co-evolution process terminates once both the refined labels $\hat{\mathbf{y}}^*$ and model parameters $\hat{\theta}$ converge (see Section 3.2).

Given the space limit, readers can refer to the Appendix A for the dictionary of symbols.

## 3.1 REFERENCE AUGMENTATION

Our insight lies in that noisy samples are likely to generate strong conflicting training signals with the ground-truth references, and clean samples can generate aligning or negligible training signals.

Technically, given a pre-trained model $f_\theta$ from masked image modeling, we append a fully connected classification head $g_\phi(.) : \mathbb{R}^h \to \mathbb{R}^C$ parameterized by $\phi$, which is fine-tuned on $\mathcal{D}_{\text{ref}}$. The layer is used to measure the agreement between a target training sample $\mathbf{s}_i = (\mathbf{x}_i, \tilde{\mathbf{y}}_i) \in \tilde{\mathcal{D}}$ and any reference samples $\mathbf{s}_{ref} = (\mathbf{x}_{ref}, \mathbf{y}_{ref}) \in \mathcal{D}_{ref}$. In this work, we adopt TraceIn (Pruthi et al., 2020), a practical influence function measuring how likely fitting a training sample $\mathbf{s}_i$ is helpful or harmful to predict a reference sample $\mathbf{s}_{ref}$.

$$IF(\mathbf{s}_i, \mathbf{s}_{\text{ref}}) = \frac{\nabla_\phi \mathcal{L}_i^\top \nabla_\phi \mathcal{L}_{\text{ref}}}{\|\nabla_\phi \mathcal{L}_i\| \cdot \|\nabla_\phi \mathcal{L}_{\text{ref}}\|}, \quad \text{where} \quad \begin{cases} \mathcal{L}_i = \mathcal{L}(g_\phi \circ f_\theta(\mathbf{x}_i), \tilde{\mathbf{y}}_i) \\ \mathcal{L}_{\text{ref}} = \mathcal{L}(g_\phi \circ f_\theta(\mathbf{x}_{\text{ref}}), \mathbf{y}_{\text{ref}}) \end{cases} \tag{5}$$

$$IF(\mathbf{s}_i, \mathcal{D}_{\text{ref}}) = \frac{1}{T \cdot |\mathcal{M}_i|} \sum_{t=1}^{T} \sum_{\mathbf{s}_{\text{ref}} \in \mathcal{M}_i} IF(\mathbf{s}_i, \mathbf{s}_{\text{ref}}), \quad \text{where} \quad \mathcal{M}_i = \{(\mathbf{x}_{\text{ref}}, \mathbf{y}_{\text{ref}}) \in \mathcal{D}_{\text{ref}} \mid \mathbf{y}_{\text{ref}} = \tilde{\mathbf{y}}_i\} \tag{6}$$

As shown in Equation 6, the influence of training sample $i$ is calculated as the gradient alignment between this sample and all reference samples with the same label as $\tilde{\mathbf{y}}_i$. The final influence score is averaged over $T$ training checkpoints. Training samples with positive $IF(\mathbf{s}_i, \mathcal{D}_{ref})$ larger than threshold $\delta_{IF}$ are used to construct a larger augmented reference set $\mathcal{D}^*_{ref}$. $\delta_{IF}$ is tuned to be 0.8, and we demonstrate its robustness with different settings in the Appendix F.3. The augmented reference set is further used to update $\theta$ and $\phi$, collectively referred to as $\hat{\theta}$.

## 3.2 CURATION-TRAINING CO-EVOLUTION

We formalize co-evolution as an alternating optimization scheme closely analogous to the Expectation–Maximization (EM) algorithm (Dempster et al., 1977), composed of the following two phases:

- Curation (**C**-step): Holding the current embedding estimator $\hat{f}$ fixed, we apply Equation 4 to compute refined curated labels $\hat{\mathbf{y}}^*$. In this phase, each sample is assigned a "responsibility" weight over $C$ classes, analogous to the E-step in EM algorithm (Dempster et al., 1977), i.e., posterior mixture-component assignments.
- Parameter update (**P**-step): Using the curated labels $\hat{\mathbf{y}}^*$ as targets, we update $f_\theta$ by empirical risk minimization. This update refines the embedding space, which in turn yields more accurate similarity estimation $k(\hat{f}(\mathbf{x}_{ref}), \hat{f}(\mathbf{x}_t))$ for the subsequent C-step, analogous to the M-step in EM algorithm (Dempster et al., 1977), i.e., parameter re-estimation given fixed responsibilities.

By iterating these two phases, the embedding function and the curated labels co-adapt dynamically: improved embeddings produce more reliable pseudo-labels, and more accurate pseudo-labels guide sharper embeddings, thereby ensuring progressive convergence despite the presence of label noise.

**C-step Curation Principle.** We estimate refined labels $\hat{\mathbf{y}}^*$ for all samples in the noisy dataset $\tilde{\mathcal{D}}$, based on neighborhood voting from the augmented reference set $\mathcal{D}^*_{\text{ref}}$. Given a sample $(\mathbf{x}, \tilde{\mathbf{y}})$, the estimated label distribution $\hat{\mathbf{y}}$ is defined as:

$$\hat{\mathbf{y}}(\mathbf{x}) = \frac{1}{Z(\mathbf{x})} \sum_{(\mathbf{x}_{\text{ref}}, \mathbf{y}_{\text{ref}}) \in \mathcal{D}^*_{\text{ref}}} \mathbf{y}_{\text{ref}} \cdot \mathbb{1}(\mathbf{x}_{\text{ref}} \in \mathcal{D}^*_{\text{vote}}(\mathbf{x})) \cdot \text{Cosine}(f(\mathbf{x}), f(\mathbf{x}_{\text{ref}})) \tag{7}$$

where we use cosine similarity as the kernel function, and $Z(\mathbf{x})$ is a normalization constant ensuring $\hat{\mathbf{y}}$ forms a valid probability distribution: $Z(\mathbf{x}) = \sum_{\mathbb{1}(\mathbf{x}_{\text{ref}} \in \mathcal{D}^*_{\text{vote}}(\mathbf{x}))} \text{Cosine}(f(\mathbf{x}), f(\mathbf{x}_{\text{ref}}))$.

To mitigate voting noise, we restrict the voting pool using an indicator function $\mathbb{1}(\mathbf{x}_{\text{ref}} \in \mathcal{D}^*_{\text{vote}}(\mathbf{x}))$ that enforces two criteria: (i) *semantic relevance* to the query sample, defined as

$$\mathcal{D}_{\text{vote}}(\mathbf{x}) = \{(\mathbf{x}_{\text{ref}}, \mathbf{y}_{\text{ref}}) \in \mathcal{D}_{\text{ref}}^* \mid \text{Cosine}(f(\mathbf{x}), f(\mathbf{x}_{\text{ref}})) \geq \tau\} \tag{8}$$

, where $\tau$ is set to the 75th percentile of the cosine-similarity distribution, and (ii) *inter-diversity* among selected references. For the latter, we construct a subset of $k$ samples ($k$ is set to half the pool size) that covers the distribution of $\mathcal{D}_{\text{vote}}$:

$$\mathcal{D}_{\text{vote}}^*(\mathbf{x}) = \underset{S \subseteq \mathcal{D}_{\text{vote}}(\mathbf{x}), |S|=k}{\arg\max} \sum_{i<j} \|f(\mathbf{x}_i) - f(\mathbf{x}_j)\|^2 \tag{9}$$

To sharpen the refined label distribution and emphasize confident predictions, we apply a temperature-controlled transformation following DivideMix (Li et al., 2020). Given $t \in (0, 1]$, the sharpened label is computed as $\hat{\mathbf{y}}^* = \hat{\mathbf{y}}^{1/t} / (\mathbf{1}^\top \hat{\mathbf{y}}^{1/t})$, with $t = 0.5$ in all experiments.

**P-step Optimization Objective.** Given the current model parameters $\theta$ and a curated dataset $\hat{\mathcal{D}} = (\mathbf{x}_i, \hat{\mathbf{y}}_i^*)$ obtained from the C-step, we update $\theta$ by minimizing the following objective:

$$\hat{\theta} = \underset{\theta,\phi}{\arg\min} \ \frac{1}{|\hat{\mathcal{D}}|} \sum_{i=1}^{|\hat{\mathcal{D}}|} \left[ \mathcal{L}_{\text{CE}}(g_\phi \circ f_\theta(\mathbf{x}_i), \hat{\mathbf{y}}_i^*) + \lambda \|\theta\|_2^2 \right] \tag{10}$$

Here, $\mathcal{L}_{CE}(\cdot, \cdot)$ denotes the cross-entropy loss between the model's prediction and the curated label, and $\|\theta\|_2^2$ is an $\ell_2$ regularization term that discourages overfitting and promotes smoother parameter. We provide theoretical proof of convergence on our website (TrainRef, 2025).

## 4 EXPERIMENT

We evaluate our approach with the following research questions, each addressed with an experiment:

- **RQ1 (Predictive Accuracy):** How effectively does TrainRef achieve predictive accuracy compared to state-of-the-art label denoising methods?
- **RQ2 (Confidence Reliability):** How effectively does TrainRef improve confidence calibration compared to state-of-the-art calibration methods?
- **RQ3 (User study):** In practical use, to what extent do humans agree with the confidence estimates produced by TrainRef?
- **RQ4 (Ablation study):** What are the contributions of each component to the overall performance?

Implementation details and qualitative analyses are deferred to Appendix C due to space limits.

### 4.1 RQ1: PREDICTIVE ACCURACY

**Synthetic Noisy Datasets.** We evaluate TrainRef on CIFAR-100 (Krizhevsky et al., 2009) under three common noise types: (1) *instance-dependent noise* (IDN) (Xia et al., 2020), where each instance is assigned a noise rate from a truncated Gaussian distribution with class-level rates chosen randomly; (2) *symmetric noise* (Sym.) (Li et al., 2020), where labels are flipped uniformly at random to any other class; and (3) *asymmetric noise* (Asym.), where labels are flipped to semantically similar or neighboring classes at a fixed rate. Following Li et al. (2023), we set symmetric noise levels to $\rho \in \{20\%, 50\%, 80\%\}$ and both asymmetric and instance-dependent noise to $\rho = 40\%$.

**Real-World Noisy Datasets.** To assess TrainRef in practical settings, we evaluate it on Web-Vision1.0 (Li et al., 2017) and Animal-10N (Song et al., 2019). WebVision1.0 contains 2.4M web-crawled images from Google and Flickr. Animal-10N consists of noisy labels from five pairs of visually similar animal species. Both are challenging real-world benchmarks.

**Results on CIFAR with synthetic noise.** Table 1 shows the generalization performance under various noise levels on the CIFAR-100 dataset. Overall, TrainRef consistently outperforms all the baselines, serving as a new state-of-the-art. Specifically, under severe noise setting (e.g., 80% symmetric), TrainRef surpasses L2B-C2D by over 10% and remains robust in challenging cases like asymmetric (40%) and instance-dependent (40%) noise, outperforming DISC (Li et al., 2023) by over 3%. Figure 3 provides a training example curated by TrainRef with distributional label in the instance-dependent 40% noise setting on CIFAR-100. The image belongs to the class "sea" but has been incorrectly labeled as "plain". TrainRef assigns a distributional label that balances between

Table 1: Comparison with SOTA methods on CIFAR-100 datasets with different types and levels of label noise. Mean ± standard deviation is reported over 3 runs. The results are primarily derived from (Li et al., 2023) or the original papers.

| Noise type | Sym 20% | Sym 50% | Sym 80% | Asym 40% | Inst 40% |
|---|---|---|---|---|---|
| CE | 55.17 ± 0.12 | 32.40 ± 0.16 | 7.70 ± 0.16 | 40.63 ± 0.26 | 43.17 ± 0.15 |
| Decoupling (Malach & Shalev-Shwartz, 2017) | 52.75 ± 0.11 | 27.59 ± 0.16 | 7.38 ± 0.09 | 39.12 ± 0.08 | - |
| Co-teaching (Han et al., 2018) | 51.24 ± 0.23 | 25.07 ± 0.18 | 8.50 ± 0.06 | 38.06 ± 0.15 | 23.21 ± 0.57 |
| JointOptim (Tanaka et al., 2018) | 58.50 ± 0.47 | 53.58 ± 0.43 | 24.62 ± 0.50 | 61.17 ± 0.39 | - |
| Co-teaching+ (Yu et al., 2019) | 51.24 ± 0.23 | 25.07 ± 0.18 | 9.50 ± 0.08 | 36.58 ± 0.16 | 24.45 ± 0.71 |
| GCE (Zhang & Sabuncu, 2018) | 76.16 ± 0.11 | 72.84 ± 0.12 | 28.40 ± 0.06 | 46.08 ± 0.20 | 45.69 ± 0.14 |
| PENCIL (Yi & Wu, 2019) | 55.17 ± 0.12 | 37.12 ± 0.17 | 9.33 ± 0.33 | 40.63 ± 0.26 | - |
| JoCoR (Wei et al., 2020) | 54.70 ± 0.08 | 26.45 ± 0.13 | 7.35 ± 0.05 | 37.09 ± 0.09 | 23.95 ± 0.44 |
| DivideMix (Li et al., 2020) | 76.16 ± 0.11 | 72.84 ± 0.12 | 28.40 ± 0.06 | 55.56 ± 0.53 | 76.08 ± 0.35 |
| ELR (Liu et al., 2020) | 69.93 ± 0.14 | 58.10 ± 0.17 | 28.40 ± 0.06 | 46.08 ± 0.20 | - |
| ELR+ (Liu et al., 2020) | 76.94 ± 0.18 | 73.01 ± 0.14 | 58.01 ± 0.17 | 74.39 ± 0.17 | - |
| Co-learning (Wei et al., 2020) | 69.93 ± 0.14 | 58.10 ± 0.17 | 41.77 ± 0.32 | 51.50 ± 0.24 | - |
| DISC (Li et al., 2023) | 78.75 ± 0.13 | 75.21 ± 0.15 | 57.61 ± 0.29 | 76.50 ± 0.15 | 78.44 ± 0.19 |
| L2B-C2D (Zhou et al., 2024) | 79.67 ± 0.14 | 78.23 ± 0.16 | 69.66 ± 0.19 | 78.22 ± 0.14 | 79.43 ± 0.17 |
| **Ours** | **85.44 ± 0.21** | **82.07 ± 0.17** | **77.85 ± 0.35** | **79.67 ± 0.22** | **82.33 ± 0.16** |

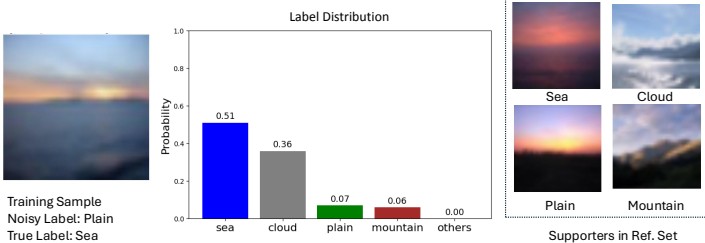

Figure 3: A noisy training sample (in the left) is curated by TrainRef to have a class distribution where the probability over the class "sea", "cloud", "plain", "mountain", and etc (in the middle). In addition, the reference samples voting for such a curation is shown in the right.

"sea" and "cloud", reflecting the inherent ambiguity in the image. In addition, we show reference samples voting this label, explaining how TrainRef makes such a curation decision. More examples are available at our website (TrainRef, 2025).

**Results on WebVision and Animal-10N.** Tables 2 and Tables 3 present the generalization performance of TrainRef with SOTA methods on real-world noisy datasets, WebVision and Animal-10N, respectively. The results demonstrate that TrainRef improves the predictive accuracy by 2% - 4% on both benchmarks, highlighting its effectiveness in handling real-world label noise.

## 4.2 RQ2: CONFIDENCE RELIABILITY

**Setup.** We evaluate calibration on CIFAR-100 under three noise regimes: noise-free, symmetric 20%, and symmetric 80%. Baseline calibration methods include (i) *Raw cross-entropy (CE) loss*, (ii) *Focal variants loss* (Focal, AdaFocal, DualFocal/AdaDualFocal (Mukhoti et al., 2020a; Ghosh et al., 2022; Tao et al., 2023)), (iii) *Post-hoc calibration* such as Temperature Scaling (TS) (Guo et al., 2017), PTS (Tomani et al., 2022), Spline (Gupta et al., 2020), MnM (Zhang et al., 2020)), (iv) *Denoising methods* (DISC, L2B) with and without TS, and TrainRef with and without TS. We report accuracy, ECE (Nixon et al., 2019), and AdaECE (Mukhoti et al., 2020b). Extended results with additional metrics ($ECE_{debias}$, $ECE_{sweep}$ (Roelofs et al., 2022)), other CIFAR-100 noise settings, and two real-world datasets (Animal-10N, WebVision) are provided in the Appendix D.

**Results.** Table 4 shows that TrainRef consistently achieves the best trade-off between accuracy and calibration across all noise settings. Focal variants losses amplify noise by up-weighting mislabeled "hard" samples, while post-hoc methods such as TS only rescale logits $\hat{p}_k = \text{softmax}(z_k/T)$ and cannot fix incorrect confidence rankings, yielding low ECE but poor accuracy. Denoising methods such as DISC and L2B improve accuracy through filtering or hard relabeling, but discard uncertain yet informative samples, thereby weakening calibration. By contrast, TrainRef utilizes soft labels

Table 2: Prediction Accuracy on WebVision.

| Accuracy (%) | Top-1 | Top-5 |
|---|---|---|
| F-correction (Patrini et al., 2017) | 61.12 | 82.68 |
| MentorNet (Jiang et al., 2018) | 63.00 | 81.40 |
| Co-teaching (Han et al., 2018) | 63.58 | 85.20 |
| ELR (Liu et al., 2020) | 76.26 | 91.26 |
| DivideMix (Li et al., 2020) | 77.32 | 91.64 |
| ELR+ (Liu et al., 2020) | 77.78 | 91.68 |
| GJS (Englesson & Azizpour, 2021) | 77.99 | 90.62 |
| CC (Zhao et al., 2022) | 79.36 | 93.64 |
| DISC (Li et al., 2023) | 80.28 | 92.28 |
| LSL (Kim et al., 2024) | 81.40 | 93.00 |
| **Ours** | **82.28** | **95.14** |
| **Ours (IN1k Pretrained)** | **84.10** | **96.34** |

Table 3: Prediction Accuracy on Animal-10N.

| Method | Accuracy (%) |
|---|---|
| CE (Englesson & Azizpour, 2021) | $79.4 \pm 0.14$ |
| GCE (Zhang & Sabuncu, 2018) | $81.5 \pm 0.08$ |
| SELFIE (Song et al., 2019) | $81.8 \pm 0.09$ |
| Mixup (Zhang, 2017) | $82.7 \pm 0.03$ |
| Co-learning (Tan et al., 2021) | 83.0 |
| PLC (Zhang et al., 2021) | $83.4 \pm 0.43$ |
| Nested Co-teaching (Chen et al., 2021b) | $84.1 \pm 0.10$ |
| GJS (Englesson & Azizpour, 2021) | $84.2 \pm 0.07$ |
| DISC (Li et al., 2023) | $87.1 \pm 0.15$ |
| LSL (Kim et al., 2024) | 89.1 |
| **Ours** | $\mathbf{90.90 \pm 0.24}$ |
| **Ours (IN1k Pretrained)** | $\mathbf{93.72 \pm 0.15}$ |

Table 4: Comparison of confidence calibration performance across noise-free, symmetric 20% (Sym20), and symmetric 80% (Sym80) settings on CIFAR-100. Results report test accuracy (higher is better) and calibration errors (ECE, AdaECE; lower is better). Grey-shaded rows indicate methods where temperature scaling (TS) is applied on top of the base method. Bold entries mark the best results under the same TS setting (either with TS or without TS).

| Method | Noise-Free | | | Sym20 | | | Sym80 | | |
|---|---|---|---|---|---|---|---|---|---|
| | Test Acc (%) | ECE (↓) | AdaECE (↓) | Test Acc (%) | ECE (↓) | AdaECE (↓) | Test Acc (%) | ECE (↓) | AdaECE (↓) |
| CE (Baseline) | 77.87 | 0.1512 | 0.1508 | 51.76 | 0.088 | 0.0879 | 16.38 | 0.0946 | 0.0946 |
| | | | | Runtime Method | | | | | |
| Focal Loss (Mukhoti et al., 2020a) | 78.31 | 0.0864 | 0.0866 | 52.16 | 0.1199 | 0.1198 | 16.26 | 0.1055 | 0.1055 |
| Ada Focal Loss (Ghosh et al., 2022) | 78.55 | 0.0723 | 0.0717 | 51.69 | 0.0923 | 0.0913 | 16.68 | 0.105 | 0.1049 |
| Dual Focal Loss (Tao et al., 2023) | 77.93 | 0.0925 | 0.0924 | 47.32 | 0.1476 | 0.1476 | 16.95 | 0.1057 | 0.1055 |
| | | | | CE + Posthoc | | | | | |
| CE+TS (Guo et al., 2017) | 77.87 | 0.0293 | 0.0297 | 51.76 | **0.0137** | **0.0138** | 16.38 | 0.0136 | 0.0097 |
| CE+PTS (Tomani et al., 2022) | 77.87 | 0.0254 | 0.0266 | 51.76 | 0.0263 | 0.028 | 16.38 | 0.014 | 0.0135 |
| CE+Spline (Gupta et al., 2020) | 77.87 | 0.0306 | 0.0331 | 51.76 | 0.0242 | 0.028 | 16.38 | 0.024 | 0.0286 |
| CE+MnM (Zhang et al., 2020) | 77.87 | 0.0212 | 0.0201 | 51.76 | 0.0177 | 0.0126 | 16.38 | 0.0134 | **0.0085** |
| | | | | Mixture | | | | | |
| DISC (Li et al., 2023) | $81.23 \pm 0.10$ | $0.113 \pm 0.013$ | $0.112 \pm 0.011$ | $78.75 \pm 0.13$ | $0.118 \pm 0.011$ | $0.114 \pm 0.016$ | $57.61 \pm 0.29$ | $0.12 \pm 0.013$ | $0.147 \pm 0.016$ |
| DISC+TS | $81.23 \pm 0.10$ | $0.025 \pm 0.007$ | $0.027 \pm 0.007$ | $78.75 \pm 0.13$ | $0.043 \pm 0.005$ | $0.045 \pm 0.010$ | $57.61 \pm 0.29$ | $0.061 \pm 0.007$ | $0.053 \pm 0.012$ |
| L2B (Zhou et al., 2024) | $82.31 \pm 0.14$ | $0.124 \pm 0.011$ | $0.131 \pm 0.009$ | $79.67 \pm 0.14$ | $0.103 \pm 0.013$ | $0.112 \pm 0.009$ | $69.66 \pm 0.19$ | $0.133 \pm 0.009$ | $0.152 \pm 0.022$ |
| L2B+TS | $82.31 \pm 0.14$ | $0.027 \pm 0.008$ | $0.028 \pm 0.009$ | $79.67 \pm 0.14$ | $0.042 \pm 0.012$ | $0.043 \pm 0.011$ | $69.66 \pm 0.19$ | $0.057 \pm 0.015$ | $0.061 \pm 0.017$ |
| Ours | $\mathbf{85.87 \pm 0.15}$ | $\mathbf{0.041 \pm 0.008}$ | $\mathbf{0.043 \pm 0.010}$ | $\mathbf{85.44 \pm 0.21}$ | $\mathbf{0.048 \pm 0.009}$ | $\mathbf{0.047 \pm 0.008}$ | $\mathbf{77.85 \pm 0.35}$ | $\mathbf{0.082 \pm 0.013}$ | $\mathbf{0.086 \pm 0.011}$ |
| Ours+TS | $\mathbf{85.87 \pm 0.15}$ | $\mathbf{0.015 \pm 0.007}$ | $\mathbf{0.014 \pm 0.008}$ | $\mathbf{85.44 \pm 0.21}$ | $0.015 \pm 0.009$ | $0.016 \pm 0.006$ | $\mathbf{77.85 \pm 0.35}$ | $\mathbf{0.011 \pm 0.005}$ | $0.014 \pm 0.009$ |

and a minimal reference set $\mathcal{D}_{\mathrm{ref}}$ to retain uncertainty and provide reliable supervision, resulting in stronger calibration and accuracy.

## 4.3 RQ3: USER STUDY

**Setup.** We collect confidence scores from the test set after training the models on different noise levels (i.e., 20%, 50%, and 80%) of CIFAR-100. Specifically, under a noisy rate (e.g., 20%), we learn TrainRef and the baseline DISC. Then, we select 100 test samples (1) which are predicted to have low confidence by either TrainRef or DISC; or (2) where TrainRef and DISC have a large disagreement on their confidence We hire 5 experts as participants, each with over 3 years of experience in model training and data labeling. Each participant is presented with two anonymous predictions (TrainRef or the baseline) and asked to choose a predicted confidence to agree with.

**Results.** Table 5 shows that across all noise settings, participants consistently prefer the predictions from TrainRef by a significant margin. As the noise rate increases, the preference for TrainRef becomes even more pronounced. Notably, at an 80% noise rate, in over 75% of cases, participants consider TrainRef 's predictions to be more reliable, showing its advantage to produce semantically meaningful and robust classifications under extreme noise conditions. More examples and results are available at our website (TrainRef, 2025).

## 4.4 RQ4: ABLATION STUDY

**Effectiveness of Initial Reference Set Size.** Table 6 shows the effect of varying the initial reference set size $|D_{ref}|$ on the noise rate of the augmented set $D_{ref}^*$ and test accuracy. Results are reported on CIFAR-100 with 80% symmetric noise and on two real-world datasets (Animals-10N and WebVision), for which only test accuracy is available due to the lack of clean labels. TrainRef consistently maintains a low noise rate ($< 5\%$) in the augmented reference set and high test accuracy across varying

Table 5: User Study Evaluation of Prediction Reliability: At each noise rate, the participants choose to agree with the predicted confidence of the test sample by either TrainRef or DISC.

| Sym. Noise Rate | TrainRef (%) | DISC (%) | Total (%) |
|---|---|---|---|
| 20% | 62% | 38% | 100% |
| 50% | 74% | 26% | 100% |
| 80% | 78% | 22% | 100% |

Table 6: Noise Rate and Sample Counts vs Initial Reference Set Size

| Dataset & Noise | Metric | k=1 | k=5 | k=10 | k=100 |
|---|---|---|---|---|---|
| **CIFAR-100** (Sym 80%) | CleanNum | 6069 | 6352 | 6389 | 6409 |
| | Mis | 140 | 110 | 103 | 101 |
| | NR($\downarrow$) | 0.0230 | 0.0173 | 0.0161 | 0.0157 |
| | Test Acc | 77.85 | 77.89 | 77.93 | 77.91 |
| **WebVision** | Test Acc | 82.21 | 82.28 | 82.27 | 82.36 |
| **Animals-10N** | Test Acc | 90.75 | 90.90 | 90.97 | 90.88 |

Table 7: Effectiveness of Label Distribution on Calibration (CIFAR100 Inst. 40%)

| Method | Hard Relabel | Soft Relabel | NNL $\downarrow$ | ECE $\downarrow$ | Test Acc (%) |
|---|---|---|---|---|---|
| Ours | | | 0.710 | 0.055 | 81.07 |
| Ours-TC | | | 0.707 | 0.032 | – |
| Ours | ✓ | | 0.756 | 0.065 | 81.77 |
| Ours-TC | ✓ | | 0.740 | 0.032 | – |
| Ours | ✓ | ✓ | **0.683** | **0.046** | **82.33** |
| Ours-TC | ✓ | ✓ | **0.669** | **0.017** | – |

reference sizes, remaining effective even with just one reference sample per class ($|D_{ref}| = 1$). In our experiments, we use $|D_{ref}| = 5$ to balance accessibility and performance. These results demonstrate TrainRef's ability to efficiently identify clean samples under extreme noise and minimal supervision.

**Effectiveness of Label Distribution** We evaluate TrainRef on CIFAR-100 with 40% instance-dependent noise, focusing on the effect of label distribution. Specifically, we replace soft labels with one-hot labels and apply temperature scaling as in RQ2. Lower ECE (Expected Calibration Error) and NNL (Negative Log-likelihood) indicate better calibration. As shown in Table 7, TrainRef achieves the lowest ECE with and without temperature scaling, outperforming all ablations. While one-hot labels (hard relabel) yield similar test accuracy, they degrade calibration by sharpening decision boundaries and ignoring ambiguous samples.

Additional ablations on Phase III iteration counts (Appendix F.2), the role of influence-based reference augmentation (Appendix F.1), as well as analyses of computational cost (Appendix B), backbone fairness (Appendix F.5) and limitations (Appendix H) are provided in the Appendix.

## 5 RELATED WORK

**Learning with Noisy Labels (& Data Curation)** Early LNL methods (Hendrycks et al., 2018; Patrini et al., 2017) assume class-conditional noise modeled by a label transition matrix. However, theoretical work (Chen et al., 2021a; Xia et al., 2020) shows that real-world noise is largely instance-dependent, making transition matrix estimation both inaccurate and computationally expensive. To handle instance-specific noise, prior work proposes re-weighting or filtering noisy samples using loss, confidence, or multi-view signals (Han et al., 2018; Yu et al., 2019; Li et al., 2020; Zhu et al., 2021; Kim et al., 2024). Semi-supervised learning (SSL) approaches (Sohn et al., 2020; Li et al., 2023) assign pseudo-labels to noisy samples, often relying on augmentations (Nishi et al., 2021; Cubuk et al., 2020). However, these heuristics degrade under high noise, mislabeling ambiguous samples and harming generalization (Das & Sanghavi, 2023). Meta-learning methods (Wu et al., 2021; Li et al., 2019) use clean references for guidance but incur high computational cost due to bi-level optimization.

**Confidence Calibration.** Confidence calibration aims to align predicted probabilities with true correctness likelihoods. Classical post-hoc methods include Temperature Scaling (TS) (Guo et al., 2017), parameterized transformations such as PTS (Tomani et al., 2022), spline-based mappings (Gupta et al., 2020), and Mix-n-Match (MnM) (Zhang et al., 2020). These methods adjust output probabilities after training, but they cannot fix mis-ordered confidence rankings learned under noise. Train-time calibration has also been explored: focal-style losses (Mukhoti et al., 2020a; Ghosh et al., 2022; Tao et al., 2023) emphasize hard samples but risk amplifying label noise, while denoising approaches such as DISC (Li et al., 2023) and L2B (Zhou et al., 2024) combine label correction with calibration. Evaluation metrics like ECE (Nixon et al., 2019), AdaECE (Mukhoti et al., 2020a), and recent unbiased estimators such as $ECE_{debias}$ and $ECE_{sweep}$ (Roelofs et al., 2022) provide multiple perspectives on calibration quality. Our work builds upon these foundations by integrating calibration with noise-robust training via reference-guided distributional curation.

## 6 CONCLUSION

We propose TrainRef, a training-time data curation framework that unifies label denoising and confidence calibration. Through Curation-Training Co-evolution, TrainRef refines the embedding space, maintains a diverse reference set, and assigns reliable soft labels. Extensive experiments show it outperforms state-of-the-art methods, improving accuracy and confidence calibration, with qualitative studies confirming its reliability for real-world noisy-label scenarios. In the future, we will deliver a library of TrainRef for the community and generalize the technique on generative models.

### ACKNOWLEDGMENTS

We would like to thank Koh Pang Wei from the University of Washington for his insightful discussions and inspiration. This research is supported in part by the National Natural Science Foundation of China (62572300), the Ministry of Education, Singapore (MOE-T2EP20124-0017, MOET32020-0004), the National Research Foundation, Singapore and the Cyber Security Agency under its National Cybersecurity R&D Programme (NCRP25-P04-TAICeN), DSO National Laboratories under the AI Singapore Programme (AISG Award No: AISG2-GC-2023-008-1B), and the Cyber Security Agency of Singapore under its National Cybersecurity R&D Programme and CyberSG R&D Cyber Research Programme Office. Any opinions, findings and conclusions or recommendations expressed in this material are those of the author(s) and do not reflect the views of National Research Foundation, Singapore, Cyber Security Agency of Singapore as well as CyberSG R&D Programme Office, Singapore.

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

# TECHNICAL APPENDICES AND SUPPLEMENTARY MATERIAL

## A   DICTIONARY OF SYMBOLS

Table 8: Dictionary of Symbols Used in Problem Statement (Sec. 2) and Approach (Sec. 3)

| Symbol | Description |
|---|---|
| $\tilde{D}$ | Noisy training dataset, $\tilde{D} = \{(x_i, \tilde{y}_i)\}_{i=1}^{N}$ |
| $D^*$ | Ideal clean dataset with distributional labels, $D^* = \{(x_i, y_i^*)\}$ |
| $D_{\text{ref}}$ | Manually verified small clean reference set |
| $D_{\text{ref}}^*$ | Augmented reference set derived from $D_{\text{ref}}$ |
| $\hat{D}$ | Curated dataset with refined soft labels |
| $x_i \in \mathbb{R}^d$ | Input sample in $d$-dimensional space |
| $\tilde{y}_i \in \{0,1\}^C$ | Noisy one-hot label for sample $x_i$ |
| $y_i^* \in [0,1]^C$ | Ground-truth soft label (class distribution), $\sum_{c=1}^{C} y_{ic}^* = 1$ |
| $C$ | Number of classes |
| $f_\theta : \mathbb{R}^d \to \mathbb{R}^h$ | Feature extractor / encoder with parameters $\theta$ |
| $g_\phi : \mathbb{R}^h \to \mathbb{R}^C$ | Classification head with parameters $\phi$ |
| $\hat{\theta}$ | Trained model parameters from empirical risk minimization |
| $\theta^*$ | Optimal model parameters minimizing true risk |
| $\mathcal{L}(\cdot, \cdot)$ | Loss function (e.g., cross-entropy) |
| $\lambda\|f\|$ | Regularization term (e.g., weight decay) |
| $k(x_i, x_j)$ | Similarity kernel (e.g., cosine similarity) between samples $x_i$ and $x_j$ |
| $\hat{y}(x)$ | Refined soft label of $x$ computed via voting from reference samples |
| $Z(x)$ | Normalization constant to ensure $\hat{y}(x)$ is a valid probability distribution |
| $\tau$ | Cosine similarity threshold for voting pool inclusion |
| $k$ | Number of neighbors selected for diverse reference voting |
| $D_{\text{vote}}(x)$ | Set of reference samples with similarity $\geq \tau$ to $x$ |
| $D_{\text{vote}}^*(x)$ | $k$-diverse subset of $D_{\text{vote}}(x)$ selected via max-diversity |
| $t \in (0,1]$ | Temperature parameter for sharpening predicted label distribution |
| $\text{IF}(s_i, s_{\text{ref}})$ | Influence of sample $s_i$ on $s_{\text{ref}}$ |
| $M_i$ | Subset of $D_{\text{ref}}$ with same label as $\tilde{y}_i$ |
| $\delta_{\text{IF}}$ | Threshold for influence to include sample in $D_{\text{ref}}^*$ |
| $\hat{y}^*(x)$ | Sharpened label distribution: $\hat{y}^*(x) = \hat{y}(x)^{1/t} / \sum_c \hat{y}_c(x)^{1/t}$ |

## B   COMPUTATIONAL COST ANALYSIS

**Training Efficiency.** One of the common concerns when introducing a multi-phase training framework is the potential computational overhead. In this section, we provide a detailed breakdown of the time cost of our method TRAINREF, and compare it with the top-performing baselines under the same hardware setting—specifically, a single NVIDIA GeForce RTX 4090 GPU.

**Phase-wise Training Time.** As shown in Table 9, TRAINREF comprises three phases: (1) a self-supervised pretraining phase, (2) an influence-based reference augmentation phase, and (3) a reference-guided co-evolution phase.

In **Phase I**, we apply Masked Image Modeling (MIM) using `BEiTv2` to learn a robust and generalizable embedding space. The tokenization mechanism in `BEiTv2` enables efficient training, with each MIM epoch taking only 3 minutes. We pretrain the model for 300 epochs in this stage.

In **Phase II**, we apply influence functions to augment the small trusted reference set, identifying clean samples from the noisy dataset. The model is then fine-tuned on this augmented reference

set to enhance its classification capability. This stage requires 5 fine-tuning epochs, each taking approximately 18 minutes.

In **Phase III**, we iteratively co-evolve the model and the dataset through reference-guided curation and distributional supervision. Specifically, the model refines its predictions using neighborhood voting from the reference set, while the curated dataset is simultaneously updated to reflect these refined soft labels. This iterative process ensures that both the embedding space and label quality improve progressively. The finetuning process involves 10 epochs (when $N = 2$), each taking around 18 minutes. During this stage, standard data augmentation techniques such as MixUp are applied. Thanks to the high-quality initialization from Phase I, only a small number of finetuning epochs are sufficient to achieve strong performance.

Despite incorporating a self-supervised pretraining stage, the overall runtime of TRAINREF remains comparable to the fastest baselines, demonstrating its practical efficiency.

**Overall Runtime.** As summarized in Table 9, the total training time of TRAINREF is approximately 1470 minutes, which is only marginally higher than DISC (1400 minutes), the most efficient baseline among state-of-the-art methods. Despite including a self-supervised pretraining stage, our approach remains competitive in terms of wall-clock time due to (i) the efficiency of BEiT-based MIM and (ii) the reduced number of fine-tuning epochs required.

Figure 4 further illustrates the per-epoch training time across various baselines. Notably, the runtime of TRAINREF per epoch during finetuning is comparable to that of LSL and CC. These results collectively show that TRAINREF achieves a favorable trade-off between computational cost and performance.

Table 9: Training Time Comparison on WebVision (RTX 4090)

| Method | Time per Epoch (min) | Training Epochs | Total Time (min) |
|---|---|---|---|
| CC | 23 / epoch | – | – |
| DISC | 14 / epoch | 100 | 1400 |
| LSL | 22 / epoch | 100 | 2200 |
| Ours | 3 (MIM), 18 (FT) | 300 + 15 | 1470 |

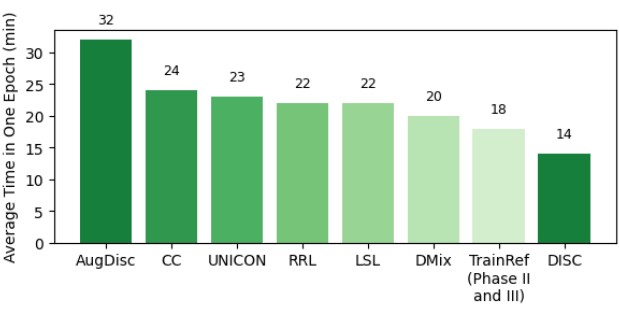

Figure 4: Training time per epoch (in minutes) across different methods. TRAINREF has a comparable finetuning cost to LSL and CC, and an efficient pretraining stage with BEiT.

## C  IMPLEMENTATION DETAILS

### C.1  MODEL ARCHITECTURE AND PRETRAINING SETUP

In Phase I of TRAINREF, we adopt the `BEiTv2` pipeline to perform self-supervised masked image modeling (MIM). The encoder is a Vision Transformer (ViT) trained from scratch on each target dataset. The model consists of 12 transformer blocks, each with 12 attention heads and a hidden dimension of 768. The patch size is set to $16 \times 16$, and input images are resized to $224 \times 224$. We use a tokenizer pretrained on ImageNet-1K, which yields 8,192 discrete visual tokens. This setup enables efficient and semantically rich representation learning, which is essential for robust downstream curation.

For MIM training, we use the AdamW optimizer with a weight decay of 0.05 and cosine annealing learning rate schedule initialized at $1 \times 10^{-3}$. A warm-up phase is applied over the first 10,000

iterations. To prevent overfitting, we use stochastic depth regularization with a drop path rate of 0.1, and stabilize optimization through layer-wise learning rate decay. Training is conducted for 200 epochs on CIFAR-100, CIFAR-80N, Animal10N, and WebVision.

## C.2 INFLUENCE-BASED REFERENCE AUGMENTATION

After pretraining, we perform linear probing to prepare for influence function analysis. A randomly initialized linear classification head is attached to the frozen encoder and trained for 15 epochs on the clean reference set. Model parameters are saved every 5 epochs to support multi-checkpoint influence estimation.
We compute the influence score of each training sample relative to reference samples using gradient similarity. Samples whose normalized influence scores exceed the threshold $\delta_{\text{IF}} = 0.8$ are selected for inclusion in the augmented reference set. By default, we initialize the reference set with 10 clean samples per class. The size and quality of this set are further analyzed in our ablation studies.

## C.3 FINETUNING AND ITERATIVE CO-EVOLUTION

In Phase II, we fine-tune the model on the augmented reference set to specialize the embedding space. We use the Adam optimizer with cosine decay, a learning rate of $1 \times 10^{-4}$, and train for 5 epochs. RandAugment is applied with parameters $(n = 2, m = 10)$ to enhance generalization, and MixUp regularization is incorporated using an interpolation coefficient of $\alpha = 0.4$.
Phase III involves two rounds of iterative co-evolution between the model and the dataset. Each iteration lasts for 5 epochs and follows the same optimization and augmentation settings as in Phase II. In each round, refined label distributions are generated via reference-guided voting, and the model is retrained on the newly curated dataset. This procedure ensures that both the embedding function and the pseudo-labels are progressively improved.

## C.4 REPRODUCIBILITY

All experiments are implemented in PyTorch and conducted on a single NVIDIA RTX 4090 GPU. Unless otherwise specified, we use a batch size of 128. Detailed training logs, configuration files, and checkpoints will be made publicly available in the project repository.

This three-phase design enables TRAINREF to efficiently extract semantically aligned embeddings, construct high-quality reference sets, and iteratively refine soft labels, ultimately yielding a robust model trained under extreme label noise.

## D EXTENDED RESULTS ON CONFIDENCE RELIABILITY

To complement the main results in Section 4.2, we provide extended evaluations on additional CIFAR-100 noise regimes (symmetric 20%, 80%, asymmetric 40%) in Table 10, 11 and 12 and on two real-world datasets (WebVision, Animal-10N) in Table 13. We also report multiple calibration metrics, including ECE, AdaECE (Mukhoti et al., 2020a), $\text{ECE}_{\text{debias}}$, and $\text{ECE}_{\text{sweep}}$ (Roelofs et al., 2022).

Table 10: Detailed results on CIFAR-100 with symmetric 20% noise.

| Method | Test Acc (%) | ECE ($\downarrow$) | AdaECE ($\downarrow$) | $\text{ECE}_{\text{debias}}$ ($\downarrow$) | $\text{ECE}_{\text{sweep}}$ ($\downarrow$) |
|---|---|---|---|---|---|
| CE | 51.76 | 0.0880 | 0.0879 | 0.0878 | 0.0880 |
| Focal Loss | 52.16 | 0.1199 | 0.1198 | 0.1197 | 0.1201 |
| Ada Focal Loss | 51.69 | 0.0923 | 0.0913 | 0.0921 | 0.0918 |
| Dual Focal Loss | 47.32 | 0.1476 | 0.1476 | 0.1474 | 0.1476 |
| CE+TS | 51.76 | **0.0137** | **0.0138** | 0.0130 | **0.0147** |
| CE+PTS | 51.76 | 0.0263 | 0.0280 | 0.0261 | 0.0261 |
| CE+Spline | 51.76 | 0.0242 | 0.0280 | 0.0240 | 0.0240 |
| CE+MnM | 51.76 | 0.0177 | 0.0126 | 0.0168 | 0.0153 |
| DISC | 78.75±0.13 | 0.118±0.011 | 0.114±0.016 | 0.117±0.009 | 0.118±0.013 |
| DISC+TS | 78.75±0.13 | 0.043±0.005 | 0.045±0.010 | 0.041±0.011 | 0.051±0.013 |
| L2B | 79.67±0.14 | 0.103±0.013 | 0.112±0.009 | 0.108±0.021 | 0.117±0.016 |
| L2B+TS | 79.67±0.14 | 0.042±0.012 | 0.043±0.011 | 0.043±0.009 | 0.045±0.012 |
| Ours | **85.44±0.21** | **0.048±0.009** | **0.047±0.008** | **0.044±0.009** | **0.052±0.014** |
| Ours+TS | **85.44±0.21** | 0.015±0.009 | 0.016±0.006 | **0.012±0.005** | 0.016±0.006 |

**Discussion.** These results confirm that TrainRef consistently outperforms state-of-the-art train-time, post-hoc, and denoising methods in both synthetic and real-world noise scenarios. Its superior

Table 11: Detailed results on CIFAR-100 with symmetric 80% noise.

| Method | Test Acc (%) | ECE ($\downarrow$) | AdaECE ($\downarrow$) | $ECE_{debias}$ ($\downarrow$) | $ECE_{sweep}$ ($\downarrow$) |
|---|---|---|---|---|---|
| CE | 16.38 | 0.0946 | 0.0946 | 0.0945 | 0.0945 |
| Focal Loss | 16.26 | 0.1055 | 0.1055 | 0.1054 | 0.1055 |
| Ada Focal Loss | 16.68 | 0.1050 | 0.1049 | 0.1048 | 0.1050 |
| Dual Focal Loss | 16.95 | 0.1057 | 0.1055 | 0.1054 | 0.1057 |
| CE+TS | 16.38 | 0.0116 | 0.0097 | **0.0069** | 0.0078 |
| CE+PTS | 16.38 | 0.0120 | 0.0135 | 0.0097 | 0.0109 |
| CE+Spline | 16.38 | 0.0240 | 0.0286 | 0.0239 | 0.0257 |
| CE+MnM | 16.38 | 0.0134 | **0.0085** | **0.0069** | **0.0068** |
| DISC | 57.61±0.29 | 0.120±0.013 | 0.147±0.016 | 0.133±0.005 | 0.154±0.015 |
| DISC+TS | 57.61±0.29 | 0.061±0.007 | 0.053±0.012 | 0.065±0.008 | 0.053±0.013 |
| L2B | 69.66±0.19 | 0.133±0.009 | 0.152±0.022 | 0.171±0.017 | 0.121±0.008 |
| L2B+TS | 69.66±0.19 | 0.057±0.015 | 0.061±0.017 | 0.055±0.011 | 0.059±0.007 |
| Ours | **77.85±0.35** | **0.082±0.013** | **0.086±0.011** | **0.080±0.007** | **0.088±0.010** |
| Ours+TS | **77.85±0.35** | **0.011±0.005** | 0.014±0.009 | 0.013±0.007 | 0.009±0.005 |

Table 12: Detailed results on CIFAR-100 with asymmetric 40% noise.

| Method | Test Acc (%) | ECE ($\downarrow$) | AdaECE ($\downarrow$) | $ECE_{debias}$ ($\downarrow$) | $ECE_{sweep}$ ($\downarrow$) |
|---|---|---|---|---|---|
| CE | 41.85 | 0.0231 | 0.0242 | 0.0228 | 0.0227 |
| Focal Loss | 38.35 | 0.0316 | 0.0319 | 0.0313 | 0.0320 |
| Ada Focal Loss | 38.71 | 0.0163 | 0.0173 | 0.0160 | 0.0151 |
| Dual Focal Loss | 32.79 | 0.0540 | 0.0556 | 0.0536 | 0.0532 |
| CE+TS | 41.85 | 0.0253 | 0.0258 | 0.0254 | 0.0260 |
| CE+PTS | 41.85 | 0.0165 | 0.0156 | 0.0162 | 0.0166 |
| CE+Spline | 41.85 | 0.0177 | **0.0188** | 0.0155 | 0.0183 |
| CE+MnM | 41.85 | 0.0235 | 0.0245 | 0.0268 | 0.0276 |
| DISC | 76.50±0.15 | 0.140±0.017 | 0.135±0.012 | 0.127±0.023 | 0.123±0.015 |
| DISC+TS | 76.50±0.15 | 0.066±0.007 | 0.061±0.009 | 0.059±0.013 | 0.057±0.009 |
| L2B | 78.22±0.14 | 0.134±0.009 | 0.121±0.011 | 0.126±0.009 | 0.142±0.011 |
| L2B+TS | 78.22±0.14 | 0.067±0.007 | 0.058±0.008 | 0.061±0.009 | 0.071±0.015 |
| Ours | **79.67±0.22** | **0.071±0.011** | **0.084±0.012** | **0.076±0.009** | **0.077±0.013** |
| Ours+TS | **79.67±0.22** | **0.015±0.005** | 0.021±0.007 | **0.014±0.006** | **0.017±0.005** |

performance stems from two principles: (i) robust anchoring via a small clean reference set, which avoids error amplification, and (ii) distributional relabeling, which preserves uncertainty while improving both accuracy and calibration.

# E  ADDITIONAL EXPERIMENTAL RESULTS ON CIFAR-80N

To further assess the robustness of TrainRef under realistic noisy-label conditions, we conduct experiments on the CIFAR-80N benchmark. Following the protocol of (Yao et al., 2021), CIFAR-80N is constructed by treating the last 20 classes of CIFAR-100 as out-of-distribution (OOD), while the remaining 80 classes are considered in-distribution. This setting introduces open-set label noise by mixing semantically unrelated classes, which challenges a model's ability to generalize under both closed-set and open-set noise.

We inject both symmetric and asymmetric label noise on the in-distribution subset, following the setup of (Sheng et al., 2024). Specifically, symmetric noise is applied at $\rho \in \{20\%, 80\%\}$ and asymmetric noise is applied at $\rho = 40\%$. These configurations allow us to evaluate model robustness under varying degrees of noise severity.

As shown in Table 14, TrainRef achieves substantial performance gains over previous state-of-the-art methods. In the Sym. 20% setting, TrainRef improves accuracy by **12.74%** over the best prior method. Under the severe Sym. 80% noise, TrainRef surpasses the closest baseline by **32.29%**. In the Asym. 40% case, which involves structured noise aligned with semantic class relationships, TrainRef achieves an improvement of **19.52%**.

These gains highlight the effectiveness of TrainRef 's unified framework in handling both closed-set and open-set noise. Notably, TrainRef does not discard OOD samples outright. Instead, it leverages reference-guided distributional labeling to assign soft targets to OOD samples based on semantic similarity. This design allows OOD instances to contribute positively to representation learning, rather than being treated as outliers.

These results reinforce the generalization ability of TrainRef in practical noisy-label scenarios, where label corruption often involves both ambiguity and distribution shift. Additional qualitative examples of TrainRef 's curation process can be found in (TrainRef, 2025).

Table 13: Accuracy and calibration on WebVision and Animal-10N. Lower calibration errors are better.

| Method | WebVision | | | | Animals-10N | | | |
|---|---|---|---|---|---|---|---|---|
| | Test Acc | ECE | AdaECE | ECE$_{debias}$ | Test Acc | ECE | AdaECE | ECE$_{sweep}$ |
| CE | 63.23 | 0.1306 | 0.1306 | 0.1287 | 80.21 | 0.1659 | 0.1656 | 0.1659 |
| DISC | 80.17 | 0.1021 | 0.1021 | 0.1008 | 87.03 | 0.0865 | 0.0865 | 0.0876 |
| Ours | **82.33** | **0.0835** | **0.0823** | **0.0819** | **90.85** | **0.0289** | **0.0282** | **0.0298** |
| CE+TS | 63.23 | 0.0277 | 0.0312 | 0.0264 | 80.21 | 0.1306 | 0.1298 | 0.1305 |
| DISC+TS | 80.17 | 0.0337 | 0.0374 | 0.0323 | 87.03 | 0.0312 | 0.0306 | 0.0350 |
| Ours+TS | **82.33** | **0.0226** | **0.0265** | **0.0213** | **90.85** | **0.0254** | **0.0221** | **0.0253** |

Table 14: Test accuracy (%) on CIFAR-80N under varying noise levels. TrainRef achieves consistent improvements across both mild and severe noise settings in open-set scenarios.

| Method | CIFAR-80N | | |
|---|---|---|---|
| | Sym. 20% | Sym. 80% | Asym. 40% |
| Standard | 29.37 | 4.20 | 22.25 |
| Co-teaching (Han et al., 2018) | 60.38 | 16.59 | 42.42 |
| Co-teaching+ (Yu et al., 2019) | 53.97 | 12.29 | 43.01 |
| JoCoR (Wei et al., 2020) | 59.99 | 12.85 | 39.37 |
| Jo-SRC (Yao et al., 2021) | 65.83 | 29.76 | 53.03 |
| SELC (Lu & He, 2022) | 57.51 | 22.79 | 47.50 |
| DivideMix (Li et al., 2020) | 57.47 | 21.18 | 37.47 |
| Co-LDL (Sun et al., 2021) | 58.81 | 24.22 | 50.69 |
| UNICON (Karim et al., 2022) | 54.50 | 36.75 | 51.50 |
| NCE (Li et al., 2022) | 58.53 | 39.34 | 56.40 |
| SOP (Liu et al., 2022) | 60.17 | 34.05 | 53.34 |
| SPRL (Shi et al., 2023) | 47.90 | 22.25 | 40.86 |
| AGCE (Zhou et al., 2023) | 60.24 | 25.39 | 44.06 |
| DISC (Li et al., 2023) | 50.33 | 38.23 | 47.63 |
| SED (Sheng et al., 2024) | 69.10 | 42.57 | 60.87 |
| **TrainRef (Ours)** | **81.84** | **74.86** | **80.39** |

# F ADDITIONAL ABLATION STUDY

## F.1 ABLATION ON INFLUENCE-BASED REFERENCE AUGMENTATION

To evaluate the effectiveness of influence-based reference set augmentation, we conduct a comparative study against several alternative strategies for reference construction and data utilization. This experiment is performed on CIFAR-100 under three distinct label noise conditions: symmetric noise at 20% and 80%, and instance-dependent noise at 40%.

We compare the following configurations:

- **KNN Embedding Voting:** Clean sample selection using $k$-nearest neighbor consistency in the embedding space, without reference set expansion or direct interaction with noisy labels.
- **Full Dataset Fine-tuning:** Standard fine-tuning on the entire noisy training set without any filtering.
- **Initial Reference Set Fine-tuning:** Model is fine-tuned only on the initial manually specified reference set (set to 500 samples).
- **First Augmented Reference Set Fine-tuning:** Model is trained using the reference set expanded via influence score-based selection.

Note that both the KNN-based method and the Initial Reference Set approach do not interact with noisy labels during training, and thus their performance remains constant across different noise configurations.

As shown in Table 15, fine-tuning on the influence-augmented reference set yields substantial gains across all noise settings. Compared to full-dataset training, the improvement exceeds 6% under symmetric 20% noise, 59% under symmetric 80% noise, and 22% under instance-dependent noise. These results underscore the importance of influence-guided augmentation in filtering out noisy examples and expanding the clean set with high precision.

Table 15: Ablation study on influence-based reference augmentation. Performance (accuracy in %) is reported under various label noise settings on CIFAR-100.

| Method | CIFAR-100 | | |
|---|---|---|---|
| | Sym. 20% | Sym. 80% | Inst. 40% |
| KNN Embedding Voting | – | 51.70 ± 1.64 | – |
| Full Dataset Fine-tuning | 66.04 ± 0.28 | 13.17 ± 1.20 | 54.83 ± 0.85 |
| Initial Reference Set Fine-tuning | – | 64.63 ± 0.18 | – |
| **1st Augmented Ref. Set Fine-tuning** | **81.24 ± 0.88** | **72.91 ± 0.73** | **76.81 ± 1.30** |

The ablation confirms that influence-based augmentation plays a central role in enabling TrainRef to scale from a minimal trusted set to a robust, curated training set, which in turn leads to substantial improvements in downstream performance.

## F.2 EMBEDDING SPACE QUALITY ACROSS PHASES

The design of TRAINREF reflects a progressive strategy to *approximate* an ideal embedding through phase-wise refinement. Our objective is to demonstrate that improved embedding quality is positively correlated with better noise detection and label refinement.

To empirically validate this, we measure the quality of the learned embedding space at each stage of the training pipeline using a non-parametric KNN classifier. Specifically, we compute the top-1 KNN classification accuracy using features extracted from the frozen encoder after each phase. The rationale is that better separation and alignment of class representations in the feature space should yield higher KNN accuracy, making it a suitable proxy for embedding quality.

Table 16: KNN classification accuracy (%) on CIFAR-100 across different phases of TRAINREF. Embedding quality improves consistently as the model progresses through the three-phase framework.

| Metric | Phase I (MIM) | Phase II (Ref. Aug) | Phase III (1st Iter) | Phase III (2nd Iter) |
|---|---|---|---|---|
| KNN Accuracy (%) | 52.18 | 75.18 | 77.78 | 79.12 |

As shown in Table 16, the embedding quality improves substantially from Phase I to Phase III. The initial self-supervised encoder achieves modest KNN accuracy (52.18%), reflecting its general-purpose nature. Fine-tuning on the influence-augmented reference set in Phase II leads to a significant jump (75.18%), and iterative refinement in Phase III further improves separability, reaching 79.12% after the second iteration.

These results empirically support our design rationale: although a perfect embedding space is not assumed, our framework steers the representation space toward that ideal through principled, iterative refinement. We will revise the main text to make this intent more explicit and to avoid any ambiguity regarding our assumptions.

## F.3 SENSITIVITY ANALYSIS OF $\delta_{IF}$

Training samples with positive $IF(\mathbf{s}_i, \mathcal{D}_{ref})$ larger than threshold $\delta_{IF}$ are used to construct an augmented reference set $\mathcal{D}_{ref}^*$. In the main experiments we set $\delta_{IF} = 0.8$, and here we study its sensitivity under different values. After constructing $\mathcal{D}_{ref}^*$, the parameters $\theta$ and $\phi$ are updated jointly, denoted $\hat{\theta}$.

Table 17 reports F1 scores on CIFAR-100 across three noise settings when varying $\delta_{IF} \in \{0.9, 0.8, 0.7\}$. The results show stable performance across different thresholds, confirming the robustness of TrainRef to the choice of $\delta_{IF}$.

## F.4 GENERALIZATION TO NON-TRANSFORMER ARCHITECTURES

To assess whether TRAINREF is limited to transformer-based architectures, we investigate its applicability to convolutional neural networks (CNNs), specifically ResNet34.

We note that Phase I of TRAINREF leverages Masked Image Modeling (MIM), which is inherently tailored to transformer-based architectures such as BEiTv2. This is because patch-level masking and reconstruction, core to MIM objectives, are not naturally compatible with the inductive biases

Table 17: Sensitivity analysis of $\delta_{IF}$ on CIFAR-100. Results are reported as F1 scores.

| CIFAR-100 Setting | $\delta_{IF} = 0.9$ | $\delta_{IF} = 0.8$ | $\delta_{IF} = 0.7$ |
|---|---|---|---|
| Sym-50% | 0.871 | 0.942 | 0.920 |
| Asym-40% | 0.834 | 0.921 | 0.907 |
| Inst-40% | 0.866 | 0.934 | 0.919 |

of CNNs. However, once the reference-guided soft labels are obtained, the curated dataset is architecture-agnostic and can be used to train alternative backbones.

To explore this, we adopt a hybrid setup where BEiTv2 is used solely for Phase I to obtain soft labels, and a ResNet34 is trained from scratch in Phases II and III using the curated dataset. Table 18 summarizes the performance under symmetric and instance-dependent label noise on CIFAR-100.

Table 18: Test accuracy (%) on CIFAR-100 with different architectures. BEiTv2 is used for soft-label generation, and ResNet34 is trained from scratch on the curated dataset. Despite underperforming the end-to-end BEiTv2 pipeline, the hybrid setup outperforms the strongest ResNet-based baseline (L2B-C2D), demonstrating architecture generalizability.

| Method (Backbone) | Sym. 50% | Sym. 80% | Inst. 40% |
|---|---|---|---|
| DISC (Li et al., 2023) (ResNet34) | $75.21 \pm 0.15$ | $57.61 \pm 0.29$ | $78.44 \pm 0.19$ |
| L2B-C2D (Zhou et al., 2024) (ResNet34) | 78.10 | 69.60 | – |
| TRAINREF (BEiTv2 → ResNet34) | $78.98 \pm 0.11$ | $74.80 \pm 0.17$ | $79.87 \pm 0.13$ |
| TRAINREF (BEiTv2 end-to-end) | $\mathbf{82.07 \pm 0.17}$ | $\mathbf{77.85 \pm 0.35}$ | $\mathbf{82.33 \pm 0.16}$ |

These results show that although using BEiTv2 end-to-end yields the strongest performance—likely due to continuity in feature learning from MIM to classification—the hybrid setup still achieves significant gains over state-of-the-art CNN-based baselines. This underscores the robustness and modularity of our reference-based relabeling framework, which can benefit downstream models regardless of architecture.

We conclude that while transformer-based architectures are preferred due to their compatibility with MIM, the relabeling and curation components of TRAINREF are generalizable and transferable to alternative backbones such as CNNs.

## F.5    FAIRNESS OF BACKBONE CHOICE

TRAINREF adopts a transformer-based backbone (BEiTv2) for its end-to-end pipeline, whereas many prior baselines are implemented with ResNet-50. To ensure that the performance gains of TRAINREF are not solely attributable to architectural differences, we re-evaluate DISC and L2B under the same transformer backbone. This provides a fair comparison by aligning backbone capacity across methods.

Table 19 reports results on CIFAR-100 (Sym. 20%, Asym. 40%), WebVision, and Animals-10N. Transformer backbones improve both DISC and L2B compared to their ResNet-50 counterparts, but TRAINREF consistently achieves the highest accuracy. This indicates that while backbone choice contributes to performance, the primary gains arise from the proposed reference-based curation framework.

Table 19: Test accuracy (%) of DISC, L2B, and TRAINREF with ResNet-50 and transformer backbones. Results show that TRAINREF's improvements persist under fair backbone alignment, confirming that the advantage is not due to architectural bias.

| Method (Backbone) | CIFAR-100 Sym. 20% | CIFAR-100 Asym. 40% | WebVision | Animals-10N |
|---|---|---|---|---|
| DISC (ResNet-50) | 78.75 | 76.50 | 80.28 | 87.10 |
| DISC (Transformer) | 80.31 | 77.52 | 80.79 | 88.45 |
| L2B (ResNet-50) | 79.67 | 78.22 | 80.56 | 89.03 |
| L2B (Transformer) | 80.91 | 79.03 | 81.15 | 89.92 |
| TRAINREF (Transformer) | **85.44** | **79.67** | **82.28** | **90.90** |

These findings demonstrate that transformer backbones provide benefits across methods, but the consistent superiority of TRAINREF highlights the effectiveness of its reference-based curation strategy rather than architectural advantage alone.

## G  PERFORMANCE UNDER NOISE-FREE CONDITIONS

To further assess the effectiveness and generalizability of TRAINREF, we report its performance under fully clean training conditions using standard cross-entropy (CE) loss. This experiment serves to answer whether the proposed soft-labeling framework is still beneficial in the absence of label noise.

We evaluate TRAINREF and several strong baselines on CIFAR-100 and CIFAR-80N under noise-free settings. Additionally, we conduct an ablation in which we disable the soft-labeling component of our method and train solely on one-hot targets derived from the clean labels.

Table 20: Test accuracy (%) on CIFAR-100 (noise-free) and CIFAR-80N (close-set noise-free, open-set noise at 20%). TRAINREF achieves state-of-the-art performance in both settings, showing benefits of soft-labeling and robustness under partial open-set corruption.

| Method | CIFAR-100 (Clean) | CIFAR-80N (20% Open-Set Noise) |
|---|---|---|
| CE (Standard Cross-Entropy) | 77.87 ± 0.17 | 64.12 ± 0.16 |
| DISC (Li et al., 2023) | 81.23 ± 0.10 | 68.88 ± 0.13 |
| SED (Sheng et al., 2024) | 67.48 ± 0.21 | 69.80 ± 0.19 |
| TRAINREF (w/o soft label) | 83.77 ± 0.10 | 80.19 ± 0.13 |
| **TRAINREF** | **85.87 ± 0.15** | **82.81 ± 0.20** |

As shown in Tables 20, TRAINREF achieves 85.87% accuracy on CIFAR-100 and 82.81% on CIFAR-80N under noise-free conditions. These results are only marginally lower than those obtained under symmetric 20% noise (85.44% and 81.84%, respectively), with performance drops of just 0.43% and 0.97%. In contrast, the best baseline (DISC) experiences significantly larger degradations of 2.48% and 8.64%, respectively.

Furthermore, removing the soft-labeling component from TRAINREF leads to noticeable declines in accuracy, even under clean supervision. This supports our claim that rigid one-hot labels may introduce inductive bias or semantic overconfidence, particularly in ambiguous instances, and that learning from distributional supervision remains beneficial.

These findings validate the utility of our approach in both noisy and clean regimes and emphasize the general-purpose benefit of soft label learning.

## H  LIMITATIONS

While TRAINREF demonstrates strong performance across noisy vision benchmarks, several limitations remain:

- **Generalization to Other Modalities.** Our study is limited to image classification tasks. Although the framework of reference-guided distributional curation is conceptually extensible, adapting it to other modalities such as text and speech requires careful design of influence functions and embedding spaces that may differ substantially from vision tasks.
- **Scalability to Large-Class Problems.** Even though TRAINREF is effective with as little as one clean sample per class, scaling to tasks with tens of thousands of classes (e.g., fine-grained clinical coding) still requires non-trivial human effort to collect a sufficiently diverse reference set. Reducing this dependency on human annotation remains an important direction.
- **Reliance on Reference Anchors.** The success of our method hinges on the availability of a trusted reference set, however small. In domains where no reliable clean data exists, alternative strategies for bootstrapping anchors are necessary.

These limitations highlight opportunities for future work, particularly in extending TRAINREF to broader modalities and reducing its reliance on human effort in extremely large-scale classification settings.

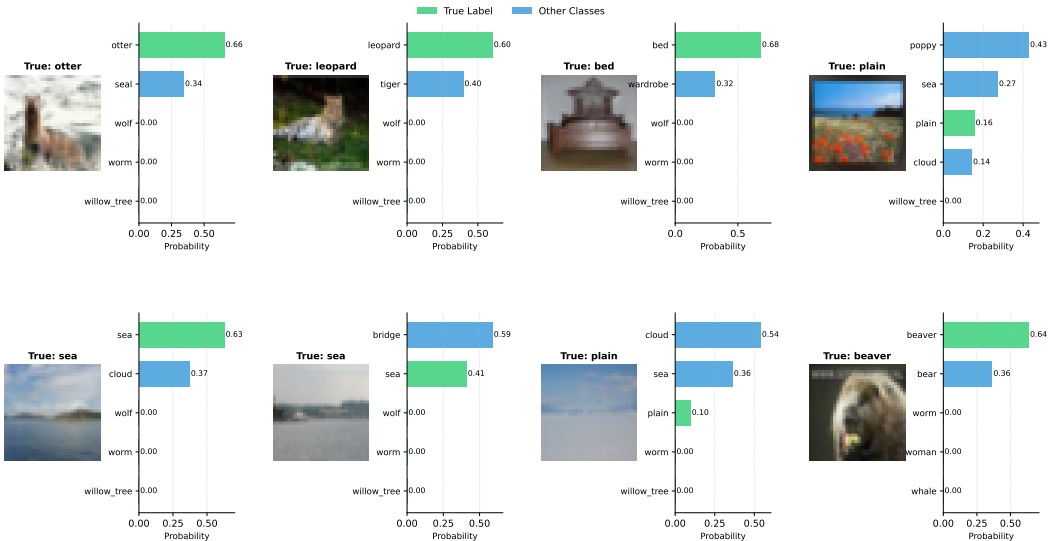

Figure 5: Representative ambiguous CIFAR-100 samples with expert-provided soft labels used to evaluate distributional-noise curation.

# I    PER-TYPE ANALYSIS OF CATEGORICAL VS. DISTRIBUTIONAL NOISE

TrainRef is designed to address two complementary forms of label misinformation: (i) *categorical noise*, where the ground-truth is one-hot but the observed label is flipped, and (ii) *distributional noise*, where the ground-truth should be a soft class distribution due to inherent ambiguity. To quantify TrainRef's effectiveness on each type separately, we conduct the following controlled analysis on CIFAR-100.

**Subset construction.**    We embed all CIFAR-100 training images using a pretrained DINOv2 encoder and compute a local neighborhood label distribution for each sample via $k$-nearest-neighbor voting in embedding space. We then use the entropy of this neighborhood distribution as an ambiguity indicator:

- **Distributional-noise subset (ambiguous).** We select samples with high neighborhood entropy ($H > 1.5$), and randomly sample 50 cases. Three independent experts annotate each case with a soft label distribution. Representative samples and expert-provided soft labels are shown in Figure 5.
- **Categorical-noise subset (unambiguous + injected flips).** We select low-entropy samples ($H < 0.1$) as unambiguous instances, inject 20% symmetric hard-flip noise, and evaluate TrainRef's ability to identify and remove mislabeled samples.

**Evaluation metrics.**    For the categorical-noise subset, we report the mislabeled fraction before and after curation. For the distributional-noise subset, we measure the KL divergence between TrainRef's curated soft labels and the expert soft labels.

**Results.**    After TrainRef curation:

- **Categorical noise rate:** $20\% \rightarrow 0.32\%$.
- **Distributional noise (KL to human soft labels):** $1.67 \rightarrow 1.43$.

These results indicate that TrainRef removes categorical noise aggressively by filtering or correcting clear label flips, while refining distributional noise more subtly by shifting labels toward calibrated soft distributions rather than discarding them. Importantly, as demonstrated in Table 7, preserving and curating distributional labels is crucial for both accuracy and confidence calibration, even when the absolute reduction in KL is smaller.

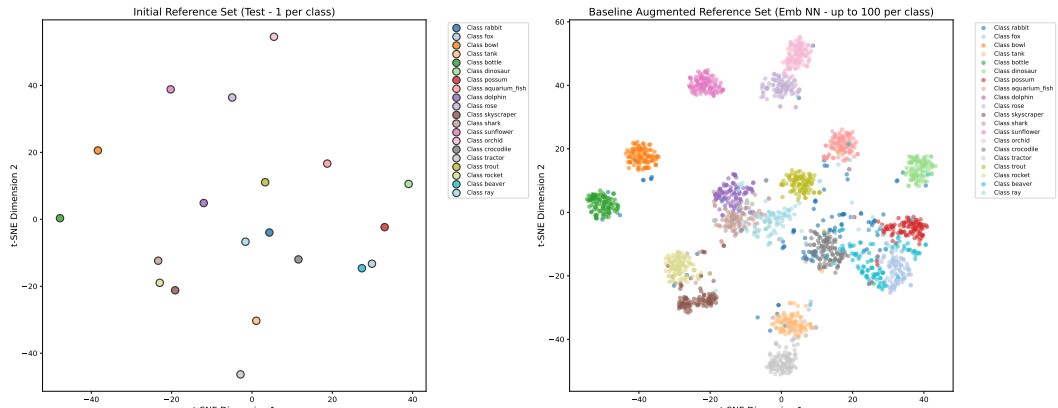

Figure 6: Embedding-NN augmentation expanded from the same initial $D_{\text{ref}}$. The expansion clusters tightly around the seeds, indicating limited diversity gain.

## J  REFERENCE-SET DIVERSITY: MEASUREMENT, AUGMENTATION BEHAVIOR, AND THRESHOLD SENSITIVITY

TrainRef relies on a small clean reference set $D_{\text{ref}}$ and its influence-augmented expansion $D_{\text{ref}}^*$. A key practical concern is whether $D_{\text{ref}}$ is sufficiently diverse to represent semantic modes within each class, and whether influence-based augmentation preserves or improves this diversity.

**Measuring diversity.**  Diversity is not characterized by set size alone.  We measure *semantic diversity within each class* using the average pairwise cosine similarity of reference embeddings:

$$\text{Sim}_{\text{intra}}(c) = \frac{2}{|D_{\text{ref}}^c|(|D_{\text{ref}}^c| - 1)} \sum_{i<j} \cos(z_i, z_j),$$

where $z_i$ is the DINOv2 embedding of sample $i$ and $D_{\text{ref}}^c$ denotes reference samples in class $c$. We report the mean over classes. Lower $\text{Sim}_{\text{intra}}$ indicates broader coverage of distinct semantic modes (higher diversity).

**Why influence augmentation does not collapse diversity.**  Influence scores are computed via *gradient alignment* with the reference training signal (Sec. 3.1), rather than raw embedding proximity. A candidate is added to $D_{\text{ref}}^*$ if it strengthens (or at least does not conflict with) the reference objective. As a result, TrainRef can select label-consistent yet embedding-diverse samples, instead of only near-duplicates of the initial seeds.

**Empirical comparison at matched size.**  To isolate the effect of augmentation strategy from reference size, we compare two expansions with the same number of added samples per class: (i) **Embedding-NN augmentation**, which adds nearest neighbors in embedding space; and (ii) **Influence augmentation (ours)**, which adds samples with high influence scores (Sec. 3.1). Average intra-class cosine similarity (lower = more diverse):
- Embedding-NN augmentation: 0.67
- Influence augmentation (ours): 0.55

Figures 6 and 7 provide qualitative evidence: embedding-NN expansion concentrates around the initial seeds, while influence expansion covers multiple semantic modes per class.

**Influence-threshold sensitivity.**  The influence threshold $\delta_{\text{IF}}$ primarily controls the *cleanliness* of $D_{\text{ref}}^*$ with an indirect cleanliness–diversity trade-off: higher $\delta_{\text{IF}}$ yields a cleaner but potentially narrower expansion, while lower $\delta_{\text{IF}}$ admits mildly aligned samples that may increase coverage but risk adding noise. Sensitivity results in Appendix F.3 (Table 17) show TrainRef remains stable across a reasonable range of $\delta_{\text{IF}}$, indicating that performance does not hinge on a finely tuned threshold.

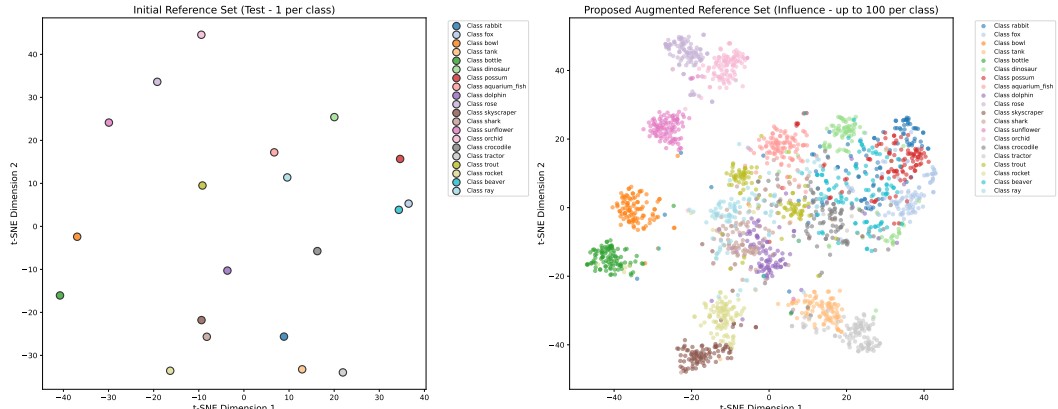

Figure 7: Influence-based augmentation expanded from the same initial $D_{\text{ref}}$. The expansion covers multiple semantic modes per class, increasing reference diversity.

## K EFFECT OF INITIAL NOISE AND INFLUENCE THRESHOLD ON PHASE III CONVERGENCE

We provide a theoretical insight into how the initial noise level of the training set ($p_0$) and the influence threshold ($\delta_{\text{IF}}$) used in Phase II affect the convergence speed of Phase III co-evolution.

**Setup recap.** Let $p_0 := \mathbb{P}[\tilde{\mathbf{y}} \neq \mathbf{y}^\star]$ be the initial label noise rate in the noisy training set $\tilde{\mathcal{D}}$. Phase II augments the clean reference set $\mathcal{D}_{\text{ref}}$ into $\mathcal{D}_{\text{ref}}^*$ by selecting training samples whose influence score exceeds $\delta_{\text{IF}}$.
Define

$$\alpha_c(\delta_{\text{IF}}) = \mathbb{P}[\text{clean sample added}], \quad \alpha_n(\delta_{\text{IF}}) = \mathbb{P}[\text{noisy sample mistakenly added}],$$

and the noise-to-clean ratio

$$\kappa(\delta_{\text{IF}}) := \frac{\alpha_n(\delta_{\text{IF}})}{\alpha_c(\delta_{\text{IF}})}.$$

Then the resulting noise rate of $\mathcal{D}_{\text{ref}}^*$ is

$$q(\delta_{\text{IF}}; p_0) = \frac{p_0 \kappa(\delta_{\text{IF}})}{(1 - p_0) + p_0 \kappa(\delta_{\text{IF}})}. \tag{11}$$

**Assumption K.1** (i.i.d. soft voting model). For a fixed sample with true label $y^\star$, let $Z_j \in [0, 1]$ denote the soft weight assigned to $y^\star$ by the $j$-th voting neighbor in the C-step. Assume $\{Z_j\}_{j=1}^K$ are i.i.d. with mean $\mu(\delta_{\text{IF}}; p_0) > \frac{1}{2}$.

**Assumption K.2** (Clean/noisy neighbor separation). There exists $\beta \in (1/2, 1]$ such that

$$\mu_c(\delta_{\text{IF}}) := \mathbb{E}[Z_j \mid j \text{ clean}] \geq \beta, \qquad \mu_n(\delta_{\text{IF}}) := \mathbb{E}[Z_j \mid j \text{ noisy}] \leq 1 - \beta.$$

**Key bound on C-step error.** Let $\bar{Z} = \frac{1}{K} \sum_{j=1}^K Z_j$ be the average soft support for the true class. By Hoeffding's inequality and Assumptions K.1–K.2,

$$\mathbb{P}[\text{C-step wrong}] = \mathbb{P}\left[\bar{Z} \leq \tfrac{1}{2}\right] \leq \exp\left(-2K(\mu - \tfrac{1}{2})^2\right)$$
$$\leq \exp\left(-c\left(1 - 2q(\delta_{\text{IF}}; p_0)\right)^2\right), \tag{12}$$

where $c := 2K(\beta - \frac{1}{2})^2$. Thus, a cleaner augmented reference set (smaller $q$) yields a smaller C-step error.

**Assumption K.3** (Co-evolution error contraction). One full co-evolution iteration contracts the classification error:

$$e_{t+1} \leq \rho(p_0, \delta_{\text{IF}}) e_t, \quad \rho(p_0, \delta_{\text{IF}}) = \exp\left(-c\left(1 - 2q(\delta_{\text{IF}}; p_0)\right)^2\right) \in (0, 1).$$

**Theorem K.4** (Iteration complexity of Phase III). *Under Assumptions K.1 and K.3, if $q(\delta_{\text{IF}}; p_0) < \frac{1}{2}$, then Phase III converges linearly:*

$$e_t \leq \rho(p_0, \delta_{\text{IF}})^t e_0.$$

*To achieve $e_t \leq \varepsilon$, it suffices to take*

$$T \geq \frac{\log(e_0/\varepsilon)}{-\log \rho(p_0, \delta_{\text{IF}})} = \frac{\log(e_0/\varepsilon)}{c\,(1 - 2q(\delta_{\text{IF}}; p_0))^2}. \tag{13}$$

*Setting $e_0 \approx p_0$ and substituting equation 11 yields*

$$T \geq \frac{\log(p_0/\varepsilon)}{c} \cdot \frac{\left((1 - p_0) + p_0\kappa(\delta_{\text{IF}})\right)^2}{\left(1 - p_0(1 + \kappa(\delta_{\text{IF}}))\right)^2}. \tag{14}$$

**Interpretation.** Equation equation 14 makes the dependence explicit:

- **Effect of initial noise $p_0$.** Larger $p_0$ increases the required iterations through both the $\log(p_0/\varepsilon)$ term and by enlarging $q(\delta_{\text{IF}}; p_0)$, which weakens the contraction factor.
- **Effect of influence threshold $\delta_{\text{IF}}$.** Increasing $\delta_{\text{IF}}$ makes Phase II selection more stringent, decreasing $\kappa(\delta_{\text{IF}})$ and thus $q(\delta_{\text{IF}}; p_0)$. This strengthens contraction and reduces $T$. Conversely, an overly low threshold may admit more noisy references, increasing $q$ and slowing convergence.

**Example.** Suppose $p_0 = 0.5$, $\delta_{\text{IF}} = 0.8$, and $\kappa(\delta_{\text{IF}}) \approx 1/20$ (i.e., clean samples are $\sim 20\times$ more likely to be selected than noisy ones). To reach $\varepsilon = 0.2$, equation 14 gives

$$T \gtrsim \frac{\log(5/2)}{c} \cdot \frac{(1 + \kappa)^2}{(1 - \kappa)^2} \approx \frac{1.12}{c},$$

suggesting that only $\sim 2$ iterations are sufficient when $c \approx 1$. Empirically, we observe that 3 Phase III iterations are enough for convergence across most noise settings, consistent with the bound.

## L USE OF LARGE LANGUAGE MODELS

In preparing this manuscript, we employed large language models (LLMs) solely as auxiliary tools for language refinement. Their usage was limited to polishing expressions, checking grammar, and improving readability. No parts of the technical content, experimental design, analysis, or results were generated by LLMs. All scientific contributions, methods, and evaluations presented in this paper were conceived, implemented, and validated entirely by the authors.

