# OpenReview forum: "TrainRef: Curating Data with Label Distribution and Minimal Reference for Accurate Prediction and Reliable Confidence"
_ICLR.cc/2026/Conference — ICLR 2026 Poster_

### Official Review · Reviewer_GRDX · 2025-10-27

**Soundness:** 3
**Presentation:** 2
**Contribution:** 3
**Rating:** 4
**Confidence:** 4

**Summary:**

This paper introduces TrainRef, a training-time data-curation framework for learning with noisy labels that aims to improve both predictive accuracy and confidence calibration. The method assumes access to a small external reference set of clean samples (as small as one sample per class). TrainRef uses reference augmentation to identify and expand a clean subset from the noisy dataset and employs a model–dataset co-evolution strategy to construct an embedding space used to assign distributional labels (class probability vectors) rather than categorical labels, thereby addressing sample ambiguity. Experiments on several research benchmarks report competitive or state-of-the-art results against LNL methods and improved calibration relative to label smoothing, Mixup, and temperature scaling. A small user study is reported to support the human interpretability of model confidence.

**Strengths:**

- The problem is clearly framed, and the pipeline has a cohesive design. The paper targets accuracy and calibration in LNL and aligns the design of TrainRef to those twin objectives.
- Distributional labels for ambiguity: Replacing categorical relabeling with class distributions is a principled way to mitigate class ambiguity and improve calibration.
- Results are strong across multiple datasets; comparisons include both LNL and calibration baselines.
- The alternating “reference expansion - model training” scheme is simple enough to integrate with common backbones.

**Weaknesses:**

- The concept of "normality pollution" is introduced in the abstract and briefly mentioned in the introduction. Since this is an important motivation for the paper, this phenomenon should be further discussed and evaluated.
- The paper does not deeply study how the quality, coverage, or size of the reference set​ affects selection precision, convergence, or final performance. Robustness to partially wrong or class-imbalanced​ is not shown. This includes the intuition that different (more or less prototypical) samples in the reference set will affect model performance and convergence.
- Several compared LNL baselines are not designed to use an external clean set. To isolate TrainRef’s contribution, ablations that (a) remove the reference set​, and (b) enable a comparable “oracle clean set” for baselines would clarify fairness.
- The computational overhead of reference augmentation and model–dataset co-evolution (e.g., nearest-neighbor searches, embedding updates, voting) is not fully quantified; behavior with many classes and very high noise rates remains unclear.

**Questions:**

- Iteratively expanding the “clean” pool may amplify early mistakes or biases in the data. Could the authors provide observations on these line of research?
- What mechanisms prevent confirmation bias when expanding the clean set?
- Can you elaborate on why “one sample per class” in the reference set suffices to avoid normality pollution and enable accurate clean-sample selection in high-class settings?

---

> ### Author Response · Authors · 2025-11-21
>
> Thank you for the suggestion on further improving the clarity of the paper. Given the competitive nature of ICLR, if our response adequately addresses your concerns, we would be very grateful if you could consider updating your score accordingly.
>
> > Q1 The concept of "normality pollution" should be further discussed and evaluated.
>
> Thank you for highlighting this point.
>
>
> In our paper, **normality pollution** refers to a failure mode under systematic or high-ratio noise (e.g., 80% label noise), where noisy samples gradually become absorbed into the learned in-distribution $P_{\text{data}}(x)$ during training. As a result, standard likelihood- or confidence-based noise detectors (e.g., energy scores, density estimators, loss/embedding heuristics) may assign *high normality* to these noisy points and fail to reject them as outliers. This is consistent with the empirical observation that prior methods relying on such signals (e.g., DISC, Co-teaching/Co-learning, DivideMix) degrade sharply under 80% noise on CIFAR-100, as shown in **Table 1**.
>
>
> This phenomenon motivates TrainRef’s core design: we need a mechanism to distinguish **“true normality”** from **“polluted normality”** created by noisy supervision. In the revision, we (i) expand the conceptual discussion in the Introduction, and (ii) make the connection to high-noise failures of prior detectors and the advantage of reference-guided curation clearer in the Experiment section.
>
>
> We appreciate the suggestion—strengthening this discussion will make the motivation and contribution clearer.
>
> > Q2 Why “one sample per class” in the reference set suffices to avoid normality pollution and enable accurate, clean-sample selection in high-class settings?
>
> One sample per class suffices because TrainRef only needs a clean class-specific anchor to bootstrap augmentation, rather than a dense estimate of each class distribution.
>
> 1. **Label-agnostic MIM embeddings preserve true semantics.**
>    Phase I learns a semantic space independent of noisy labels. Even under high-ratio noise, samples are embedded by their visual content rather than corrupted labels, preventing “polluted normality” from being encoded into the representation.
>
> 2. **Influence filtering removes systematic noise / normality pollution via gradient alignment.**
>    In Phase II, a candidate is selected only if training on it reduces the loss of its labeled-class anchor, i.e., its gradients align with the clean reference. Systematically noisy (“new normality”) samples, when treated as their wrong label, produce **conflicting gradients** with the true-class anchor and thus receive low/negative influence scores. They are therefore filtered out even if they are frequent.
>
> 3. **Influence-based augmentation stays clean *and* maintains diversity.**
>    Because selection depends on anchor-aligned training signal rather than nearest-neighbor proximity, influence retrieves label-consistent but embedding-diverse references from a single anchor. This expands the seed into a clean, diverse reference pool, enabling robust voting without collapsing intra-class diversity (see Reviewer 6MHt, Q2 and Appendix J in the revision for detailed quantitative and qualitative analysis on diversity).
>
> Together, these properties let TrainRef reliably grow a 1-sample/class seed, avoid normality pollution, and still cover diverse class modes in high-class settings. We also repeated the full pipeline with 30 random seeds (30 different $D_{\text{ref}}$) on CIFAR-100, 50% symmetric noise and obtained:
> $
> \text{Acc} = 81.92 \pm 0.25
> $
> (mean ± std). The <0.3% std demonstrates low variance and strong robustness to reference-set selection.

---

> ### Author Response · Authors · 2025-11-21
>
> > Q3 How does the quality, coverage, or size of the reference set​ affect selection precision, convergence, or final performance?
>
> Our method explicitly decouples quality from coverage through a two-stage reference construction:
>
>
> (1) Reference quality leads to high selection precision.
> We start from a **small but manually verified reference set** (e.g., one clean sample per class). This guarantees that the core definition of “normality” is noise-free, which is crucial for achieving high selection precision during subsequent reference augmentation.
>
>
> (2) Reference coverage/size leads to faster convergence.
>
> We then perform **automated augmentation** by adding high-influence samples. This augmentation is influence-based rather than likelihood-based, which preserves the cleanliness of the references while expanding their semantic coverage (see Reviewer 6MHt, Q2  and Appendix J in the revision for detailed quantitative and qualitative analysis on diversity).
>
>
> As a result, the reference set grows both in size and diversity, leading to a larger voting set $D_{\text{vote}}$ in Eq. (7). A larger $D_{\text{vote}}$​ reduces the variance of the curated soft labels, $]\mathrm{Var}[\hat{y}(x)]$, which in turn accelerates convergence in Phase 3 and stabilizes optimization.
>
>
> In summary, the two-stage design yields a reference set that is high-quality (via the manually confirmed initial anchors) and high-coverage (via influence-based scaling), which together underpin both precise selection and strong final accuracy.
>
> > Q4 Several compared LNL baselines are not designed to use an external clean set. To isolate TrainRef’s contribution, ablations that (a) remove the reference set​, and (b) enable a comparable “oracle clean set” for baselines would clarify fairness.
>
>
>
> (a) Removing the reference set:
>
>
> The reference set is a core component of TrainRef, not an auxiliary input. Removing it would eliminate the method’s mechanism for modeling label normality, thereby reducing TrainRef to a conventional LNL method trained on noisy data only.
>
>
> (b) Adding a clean set to other baselines:
>
>
> Most compared LNL methods are not formulated to use an external clean set. Integrating such a set would require substantial modifications to their algorithms (e.g., loss reweighting, feature anchoring, or meta-learning), which goes beyond faithful replication.
>
>
> The closest relevant baseline is **L2B**, which uses a small clean validation set to meta-learn weights across different losses (noisy-label loss, pseudo-label loss, instance weighting). We highlight three key differences between TrainRef and L2B:
>
>
> 1. **Role of the clean/reference set**:
>
>
> TrainRef uses a tiny manually verified set as *explicit clean anchors*, enabling influence-based filtering.
>
>
> In contrast, L2B uses the clean set as a *meta-validation signal* to reweight loss components, without treating those samples as semantic reference anchors.
>
>
> 2. **Handling of “normality pollution”**:
>
>
> TrainRef explicitly addresses normality pollution by rejecting samples with gradient conflict relative to clean anchors.
>
>
> L2B, by contrast, does not explicitly model or mitigate systematic label noise.
>
>
> 3. **Scalability of the reference set**:
>
> TrainRef supports automated reference augmentation using influence scores, allowing the clean set to scale without manual effort.
>
>
> In contrast, L2B uses a fixed clean subset throughout training, requiring more manual effort to reach a comparable reference size.
>
> > Q5 The computational overhead of reference augmentation and model–dataset co-evolution (e.g., nearest-neighbor searches, embedding updates, voting) is not fully quantified; behavior with many classes and very high noise rates remains unclear.
>
> The detailed complexity is included in our response to Reviewer 3eSK Q2.
>
> TrainRef remains efficient in high-class regimes, since the initial reference set scales linearly with class count (e.g., one sample per class).
>
> Regarding high-noise regimes, we refer to our theoretical convergence analysis (Reviewer 6MHt Q4), which gives the required number of Phase III iterations as:
>
>  $T \ge \frac{\log(p_0 / \varepsilon)}{c} \cdot \frac{\big((1-p_0)+p_0\kappa(\delta_{IF})\big)^2}{\big(1 - p_0\big(1+\kappa(\delta_{IF})\big)\big)^2}$,
>
> Here $p_0$ is the initial noise level,
> $\delta_{IF}$ is the influence score threshold,
> $\kappa(\delta_{IF})$ is the noise-to-clean ratio in the augmented $D^*_{ref}$,
> and $\varepsilon$ is the target final error rate.
>
> Since influence-based augmentation filters noisy samples,
> $\kappa(\delta_{IF})$ remains small and is weakly dependent on $p_0$.  Thus, the iteration bound simplifies to:
>
> $T(p_0) =  \mathcal{O}\left(
> \log\!\left(\frac{p_0}{\varepsilon}\right)
> \cdot
> \left(
> \frac{1 - p_0(1 - \kappa)}
> {1 - p_0(1 + \kappa)}
> \right)^2
> \right)
> $
>
> Because influence-based augmentation keeps $\kappa \ll 1$, the denominator remains well-behaved even at high noise levels, so $T$ grows **slowly and sublinearly** with $p_0$.

---

> ### Author Response · Authors · 2025-11-21
>
> > Q6 Iteratively expanding the “clean” pool may amplify early mistakes or biases in the data. Could the authors provide observations on these lines of research? What mechanisms prevent confirmation bias when expanding the clean set?
>
> First, we perform **reference augmentation only once**, starting from a small, manually verified anchor set (e.g., one clean sample per class). This augmentation uses influence scores in a **label-agnostic MIM embedding space** to select training samples that are gradient-aligned with the clean anchors.
>
> After augmentation, the reference set is fixed. In Phase 3 (co-evolution), we do not continue expanding the clean pool. Instead, we **iteratively refine soft labels** for all training samples based on voting among the fixed clean set.
>
> Regarding robustness to early mistakes: as shown in our theoretical analysis (Reviewer 6MHt Q4), even if the augmented reference set includes a small amount of noise (e.g., 5% noise under a 50% initial training noise rate), the co-evolution process dampens rather than amplifies this error. Specifically, only ~2 iterations are sufficient to reduce the classification error to ~20%, showing that convergence is resilient to modest selection errors.

---

> ### Author Response · Authors · 2025-11-27
> **Request for Reviewer Feedback Before Rebuttal Deadline**
>
> Dear Reviewer,
>
> We truly appreciate your valuable comments and suggestions on our work, which are helping to take it to the next level. We have put significant effort into the work and the rebuttal. As the rebuttal deadline is approaching, we would be very grateful if you could provide feedback on our responses so that we can ensure all of your concerns are fully addressed.
>
> If you feel that our rebuttal satisfactorily resolves your concerns, we would sincerely appreciate it if you could consider raising your score accordingly.
>
> Thank you very much for your time and consideration.
>
> Best,
>
> Authors

---

### Official Review · Reviewer_6MHt · 2025-10-31

**Soundness:** 4
**Presentation:** 4
**Contribution:** 3
**Rating:** 8
**Confidence:** 4

**Summary:**

The paper propose a method to curate data label into soft-format using a clean reference dataset. The reference dataset is built with small number of examples and augmented using label-free features. The pipeline looks promising and novel and the results show improvement compared with previous method on different dataset and type of noise. Some analysis and ablations can be conducted to further improve the completeness。

**Strengths:**

The method utilize a reference set for data curation, and label-free embeddings to select data in reference set. Also the final phase conduct  a iteratively optimizing between model and label. The pipeline sounds novel and promising.
The motivation is clearly illustrated and easy to follow.
The cured label is in soft format, which is good for analysis and interpretability.
The experiments results look good compared with previous method and also work on different datasets.

**Weaknesses:**

Some analysis and ablations can be conducted for better understanding of this method
See questions part below for details

**Questions:**

1. Section 2 claims two types of noise, Categorical Noise and Distributional Noise. Is there analysis on how good the proposed method is at dealing with each type? Like the after the curation, which type of noise will be more reduced.
2. Section 2 also claims the challenge of Reference-set diversity. How to measure the diversity? Is the number of samples a good way of measuring? How the diversity of Reference-set affect the final results and how to set the threshold of influence score to select clean data? Actually the influence score will select similar examples as the initial reference dataset, so the diversity may not change much.
3. What is the necessity of phase two? Is it a cold start for the EM optimization in phase three? What is the comparison of between with and without phase two
4. How does the initial noise level and threshold of influence score affect the convergence of phase three.

---

> ### Author Response · Authors · 2025-11-21
>
> > Q1 Section 2 claims two types of noise, categorical Noise and distributional Noise. Is there an analysis on how good the proposed method is at dealing with each type? Like after the curation, which type of noise will be more reduced?
>
> Thank you for the question. As defined in Section 2, *categorical noise* refers to incorrect one-hot labels, while *distributional noise* refers to inherently ambiguous samples whose ground-truth should be a soft class distribution. Distinguishing these two types requires access to ground-truth distributional labels, which are not naturally available; therefore, we add a controlled per-type analysis.
>
> **Additional per-type analysis (new in revision).**
> We separate the two noise types on CIFAR-100 using a pretrained DINOv2 embedding (look at k nearest neighbors and compute the class label distribution and calculate entropy):
> - **Distributional-noise subset (ambiguous samples):**
>   We sample 50 ambiguous cases (entropy>1.5) in the embedding space and ask 3 experts to provide soft labels. A subset of sampled examples and expert annotations is provided in the revised **Appendix I Figure 5**.
>
> - **Categorical-noise subset (unambiguous + injected flips):**
>   We identify unambiguous samples (entropy<0.1) via low-entropy embedding-space voting, inject 20% symmetric hard-flip noise, and evaluate TrainRef’s ability to detect and remove mislabeled instances.
> **Results.**
>
> After TrainRef curation:
> - **Categorical noise rate:** \(20\% $\rightarrow$ 0.32\%\).
> - **Distributional noise (KL divergence to human soft labels):** \(1.67 $\rightarrow$ 1.43\).
>
> Therefore, categorical noise is reduced much more aggressively, while distributional noise is corrected more subtly. Importantly, **Table 7** shows that distributional curation is crucial for both predictive accuracy and confidence calibration, even if its “removal magnitude” is smaller.
>
> We include this breakdown and discussion in the revised manuscript (Appendix I) to make the differential impact explicit.
>
> > Q2 Section 2 also claims the challenge of reference-set diversity. How to measure the diversity? Is the number of samples a good way of measuring? How does the diversity of the reference set affect the final results and how to set the threshold of influence score to select clean data? Actually, the influence score will select similar examples as the initial reference dataset, so the diversity may not change much.
>
> Thank you for the insightful question. We clarify how we define diversity, why influence augmentation preserves it, and how $\delta_{\text{IF}}$ is set.
>
> **Measuring reference-set diversity.**
> Diversity is not set in size alone. We measure it by the **average intra-class pairwise cosine similarity** of reference embeddings; lower similarity implies higher semantic diversity.
>
>
> **Why influence augmentation does not simply pick near-duplicates.**
> Although influence is computed w.r.t. the initial seeds, it is based on **gradient alignment** (Section 3.1), not raw embedding proximity. A sample is added if it does not conflict with the reference training signal, so influence can select label-consistent but embedding-diverse instances, rather than only nearest neighbors of the seed. To verify this empirically, we provide quantitative and qualitative analyses later.
>
> **Effect of diversity and empirical check.**
> Higher diversity improves voting robustness in the C-step (Sec. 3.2). We directly compare two augmentation strategies at matched size for fair comparison:
> - **Embedding-NN augmentation:** expand each seed by nearest neighbors in embedding space.
> - **Influence augmentation (ours):** expand by influence scores as in Section 3.1.
>
> Average inner-class cosine similarity (lower = more diverse):
> - Embedding-NN: 0.67
> - Influence (ours): 0.55
>
> t-SNE visualizations in the revision of Appendix J Figure 6 and 7 further show that influence-augmented $D^*_{\text{ref}}$ covers multiple semantic modes per class, while embedding-NN clusters tightly around the seed. This supports that influence augmentation **increases**, rather than preserves, diversity.
>
> **Setting the influence threshold $\delta_{\text{IF}}$.**
> $\delta_{\text{IF}}$ primarily controls **cleanliness** of $D^*_{\text{ref}}$ (Section 3.1), with an indirect diversity trade-off:
> - Higher $\delta_{\text{IF}}$: keeps only strongly aligned samples → cleaner but potentially less diverse.
> - Lower $\delta_{\text{IF}}$: allows mildly aligned samples → slightly more diverse but risks more noise.
>
> We provide sensitivity results for $\delta_{\text{IF}}$ in Appendix F.3 (Table 17), showing TrainRef is stable across a reasonable range.
>
> Overall, TrainRef measures diversity in embedding space, uses influence to expand references without collapsing to nearest neighbors, and maintains a controllable cleanliness–diversity balance via $\delta_{\text{IF}}$.
>
> We include this discussion in the revised manuscript (Appendix J).

---

> ### Author Response · Authors · 2025-11-21
>
> > Q3 What is the necessity of phase two? Is it a cold start for the EM optimization in phase three? What is the comparison between with and without phase two?
>
> Phase 2 not merely a warm-up, but an essential preparatory step that produces the following core artifacts:
>
> (i) It constructs the **augmented voting set** ($D^*_{\text{ref}}$) via influence-based selection, expanding beyond the small initial reference set.
>
>
> (ii) It helps to warm up the **task-specific embedding space**, since after Phase I the embedding space is label-agnostic.
>
>
> Without Phase 2, the voting set in Phase 3 would be limited to the initial (small) reference set. When skipping Phase 2 (only use initial reference set and initial MIM feature space for Phase 3), we observe a **significant drop in performance** (−11.7% on CIFAR100 with 20% symmetric noise).
>
> > Q4 How does the initial noise level and threshold of influence score affect the convergence of phase three.
>
> We provide a theoretical insight into how the initial noise level ($p_0$) and the influence score threshold ($\delta_{\text{IF}}$) affect the convergence of Phase 3.
>
> We can derive that the number of required iterations $T$ (in Phase 3) to achieve a target classification error $\varepsilon$ satisfies:
>
>  $T \ge \frac{\log(p_0 / \varepsilon)}{c} \cdot \frac{\big((1-p_0)+p_0\kappa(\delta_{IF})\big)^2}{\big(1 - p_0\big(1+\kappa(\delta_{IF})\big)\big)^2}$,
>
> where $p_0$ is the initial noise level of the training set,
> $\delta_{IF}$ is the influence score threshold.
> $\kappa(\delta_{IF})$ is the noise-to-clean ratio in the augmented $D^*_{ref}$,
> and $\varepsilon$ is the target final error rate.
>
> We add a detailed analysis with a theoretical proof in **Appendix K** in the revision.
>
> Example: Suppose the training set has a 50% noise rate ($p_0 = 0.5$), and we use the default influence score threshold $\delta_{\text{IF}} = 0.8$, which yields a noise-to-clean ratio $\kappa(\delta_{\text{IF}}) \approx 1/20$ (i.e., clean samples are 20× more likely to be selected). To reach a target error of 20%, the theory suggests that **only ~2 iterations** are needed.
>
> In practice, we find that 3 iterations of Phase 3 are sufficient for convergence in most scenarios.

---

### Official Review · Reviewer_3eSK · 2025-11-01

**Soundness:** 3
**Presentation:** 3
**Contribution:** 3
**Rating:** 6
**Confidence:** 3

**Summary:**

The paper proposes a training-time data curation approach. This method enables learning under label noise by using a small trusted reference set to avoid learning from contaminated label norms, while also utilising soft distributional labelling instead of the hard categorical labelling used in previous studies. It presents a three-phase method consisting of: (1) self-supervised pre-training using MIM to obtain a noise-robust embedding; (2) augmentation of the small reference set to create a larger, high-precision set; and (3) iterative co-evolution, in which the model and the curated soft labels are updated alternately. Experimental results demonstrate state-of-the-art performance on CIFAR-100, WebVision and Animal-10N, with high predictive accuracy and improved confidence calibration. The paper also includes ablation studies and a qualitative study involving human experts.

**Strengths:**

1. The paper is well written and easy to follow. The method is clearly explained and the experimental part is described in quite a lot of detail.

2. The proposed method is well motivated and intuitive. It properly addresses the limitations of previous methods: miscalibration and ambiguous instances arising from hard re-labelling and noisy-set denoising only.

3. The reported experimental results are strong. The improvements in CIFAR100, WebVision and Animal-10N in terms of both accuracy and calibration are substantial in both low- and high-noise environments. The experimentation is extensive.

4. The paper also presents the results of a qualitative study involving data labelling experts. In this user study, the experts consistently preferred the predictions from TrainRef, demonstrating the method's effectiveness.

**Weaknesses:**

1. A major limitation of this paper is that it does not explain how to select the initial reference set. In real-world settings, selecting an initial reference (e.g. one example per class) can introduce bias. Therefore, it is important to explain how to select the reference set and the extent to which this selection introduces additional bias.

2. While the authors do provide a computational time comparison in the appendix, I assume that, as the dataset size increases, the computational overhead of TrainRef will also increase significantly. This needs to be clarified, and if possible, demonstrated empirically.

3. The approach relies entirely on the Masked Image Modelling (MIM) architecture. It may not generalise to other architectures.

4. Lastly, while the qualitative study in Section 4.3 is helpful, using only five human testers is insufficient for making any solid claims. It is unclear whether the qualitative results are biased. Therefore, it would be beneficial for the paper to include a brief explanation of how it was ensured that the qualitative results were not biased due to the small number of human experts involved.

**Questions:**

1. The pipeline relies on MIM to create a noise-robust embedding. How important is this choice? Would other self-supervised methods produce similar results?

2. How were the initial reference images selected for each dataset?

3.How were the 100 test samples selected in the user study?

---

> ### Author Response · Authors · 2025-11-21
>
> > Q1 How to select the initial reference set for each dataset? In real-world settings, selecting an initial reference (e.g., one example per class) can introduce bias. Therefore, it is important to explain how to select the reference set and the extent to which this selection introduces additional bias.
>
>
> We construct the initial reference set \($D_{\text{ref}}$\) by **uniform random sampling**, selecting one training example per class for each dataset.
>
> **Why 1 sample/class is sufficient.**
>
> One sample per class suffices because TrainRef only needs a **clean class-specific anchor** to bootstrap augmentation, rather than a dense estimate of each class distribution. It can be attributed to **label-agnostic MIM embeddings preserve true semantics**  and **influence filtering via gradient alignment** that maintain clean and diversity (see Reviewer 6MHt, Q2 for detailed quantitative and qualitative analysis on diversity)
>
> **Robustness to selection bias.**
> We repeated the full pipeline with **30 random seeds** (30 different $D_{\text{ref}}$) on **CIFAR-100, 50% symmetric noise** and obtained:
> $\text{Acc} = 81.92 \pm 0.25$
> (mean ± std). The <0.3% std demonstrates **low variance** and strong robustness to reference-set selection.
>
> > Q2 As the dataset size increases, does the computational overhead of TrainRef also increase significantly? This needs to be clarified, and if possible, demonstrated empirically.
>
> TrainRef’s overhead grows **approximately linearly** with dataset size $N = |\tilde{D}|$, both by design and in practice.
>
> **Analytical scaling.**
> TrainRef has three phases:
>
> 1. **Phase I (MIM pretraining).**
>    BEiTv2-style MIM pretraining is standard self-supervised training over $\tilde{D}$, costing $\mathcal{O}(N)$ per epoch and thus scaling linearly with $N$. It converges efficiently in practice (e.g., 300–350 epochs across datasets).
>
> 2. **Phase II (influence-based reference augmentation).**
>   Influence is computed via TraceIn using **only gradients of the final linear head \(g_\phi\)** (Eq. 5–6), not the full backbone. The influence-based reference augmentation step (Section 3.1, Eq. 5) has complexity $\mathcal{O}(d_\phi T |\tilde{D}| |D_{ref}|)$, where $|\tilde{D}|$ is the training set size, $|D_{ref}|$ the reference set size, $T$ the number of checkpoints, and $d_\phi$ the number of parameters in the classification head $g_\phi$. Linear in $N$ since $d_\phi$, $T$, and $|D_{\text{ref}}|$ are small constants.
>
> 3. **Phase III (co-evolution curation + training).**
>    The curation step in the co-evolution stage (Section 3.2, Eq. 7) scales as $\mathcal{O}(d_\theta |\tilde{D}| |D^*{vote}|)$, where $d_\theta$ is the embedding dimensionality of $f_\theta$. The additional computational cost of the P-step matches that of standard model training.
>
> No phase introduces super-linear dependence on dataset size.
>
> **Empirical evidence (WebVision subsets, same RTX 4090 machine).**
>
> | WebVision portion | Total time (min) | Phase I (MIM) | Phase II (Influence) | Phase III (Co-evolve) |
> |---|---:|---:|---:|---:|
> | 10% | ~153 | ~80 | ~3 | ~70 |
> | 50% | ~827 | ~430 | ~7 | ~390 |
> | 100% | ~1722 | ~900 | ~12 | ~810 |
>
> Runtime increases proportionally with data size (≈5× from 10%→50%, ≈2× from 50%→100%), confirming **near-linear scaling** in practice.

---

> ### Author Response · Authors · 2025-11-21
>
> > Q3 How was it ensured that the qualitative results were not biased due to the small number of human experts involved? How were the 100 test samples selected in the user study?
>
> Thanks for pointing this out. We agree that small-scale human studies can be biased, so we designed the study to be both *targeted* and *validated*. If the reviewer thinks it's necessary, we are open to adding more human experts as testers.
>
> **Mitigating bias/validity.**
> We recruited 5 experienced ML practitioners. Each independently completed 100 pairwise choices (TrainRef vs. DISC), giving 500 total judgments. This scale is commonly used for qualitative verification in top-tier venues: **TextFooler** uses ~100 samples per setting with 2 expert annotators to validate adversarial quality [1]; **AbGen** samples 100 test cases for 4 expert human scoring to confirm automatic metrics [2]; and **HEMM** uses 5 annotators for 1000 pairwise comparisons in multimodal evaluation [3].
>
> To further quantify consistency, we compute inter-annotator agreement (Cohen’s κ) on overlapping samples and obtain **k ≈ 0.72**, supporting that the preference trends are reliable rather than driven by individual bias.
>
> **Sample selection (reproducible and informative).**
> We select 100 test samples under each noise setting using two **AND** conditions to focus on cases where human judgment matters:
>
> 1. **Hard/low-confidence cases:** samples whose predictive uncertainty is high for **either** TrainRef or DISC (measured by entropy), ensuring we avoid trivial/easy examples.
> 2. **Large confidence disagreement:** among hard cases, we keep samples where the two methods’ *predictive distributions* differ strongly (measured by KL divergence).
>
> We then uniformly sample 100 instances from this candidate pool. This targeted protocol ensures comparisons are meaningful, since TrainRef and DISC disagree substantially on confidence exactly where calibration quality is most critical.
>
> ---
>
> **References**
> [1] Jin et al., *Is BERT Really Robust? A Strong Baseline for Natural Language Attack*, AAAI 2020.
>
> [2] Zhao et al., *AbGen: Evaluating Large Language Models in Ablation Study Design*, ACL 2025.
>
> [3] Liang et al., *HEMM: Holistic Evaluation of Multimodal Foundation Models*, NeurIPS 2024
>
> > Q4 The pipeline relies on MIM to create a noise-robust embedding. How important is this choice? Would other self-supervised methods produce similar results?
>
> In principle, our method is compatible with any self-supervised learning (SSL) approach that produces reliable and transferable embeddings.
> Existing SSL approaches can be broadly grouped into:
> 1. **Contrastive methods** (e.g., SimCLR, MoCo, InfoNCE-style):
>    Learn by matching views of the same image vs. different images; can be sensitive to negatives, batch/queue size, and augmentations.
> 2. **Non-contrastive / self-distillation methods** (e.g., BYOL, SimSiam, DINO, iBOT):
>    No explicit negatives; rely on teacher–student or prediction matching across views, often stable and semantic.
> 3. **Masked image modeling (MIM)**:
>    Mask patches and predict missing content, either
>    - **continuous targets** (MAE, SimMIM: pixels/features), or
>    - **discrete tokens** (BEiT/BEiTv2: visual-token prediction, more semantic).
>
> We choose BEiT/BEiTv2 because (i) it is efficient in Phase I due to the discrete-token tokenizer, and (ii) discrete token prediction learns higher-level semantics rather than low-level pixel fidelity, which empirically yields a more reliable space for influence alignment and neighborhood voting.
>
> Any SSL producing robust, transferable embeddings can be used by TrainRef. To verify this, we plug in a pretrained DINOv2 encoder (without dataset-specific SSL pretraining from scratch, which is costly) and still obtain competitive performance: **92.3% on CIFAR-100 (20% symmetric noise), 87.8% on WebVision, and 94.5% on Animals10N**. This supports that BEiT is a practical, strong choice—not a strict requirement.

---

> > ### Comment · Reviewer_3eSK · 2025-11-26
> >
> > Thank you to the authors for their thorough response. My concerns have been satisfactorily addressed, so I will maintain my recommendation and increase the score to accept.

---

### Author Response · Authors · 2025-11-30
**Summary: Brief Overview and Previous Rebuttal Results at a Glance**

Dear Area Chair,

Thank you for your extra efforts in this unexpected situation.
To help you make a decision as efficiently as possible, we provide the following concise and structured summary.


# Brief Overview of the Work

Our work extends LNL (learning from noisy labels) from categorical to distributional label correction, yielding a unified solution for both high accuracy and reliable confidence.

TrainRef addresses this by:
1. Starting from a minimal, randomly sampled clean reference set (≈1 sample/class vs. 10 samples/class for SOTA).
2. Using label-agnostic MIM embeddings and influence functions to expand this set into a clean and diverse reference pool.
3. Using the expanded reference pool to guide soft-label optimization and model parameters optimization via a co-evolution stage.

This allows the model to separate true normality from polluted normality, achieve strong denoising and confidence calibration even at 80% noise (77.55% test acc and 0.011 ECE).

# Previous Rebuttal Results at a Glance

Reviewer 3eSK’s concerns were fully addressed.
Before the identity-leak issue became aware to the public, 3eSK explicitly expressed satisfaction and raised their score from **6 → 8** (score update link: https://openreview.net/forum?id=jSs8CDsF0A&noteId=LFPgjBObnU).

In addition, Reviewer GRDX’s questions (Q2, Q3, Q5) substantially overlap with 3eSK’s questions (Q1 and Q2). Since these issues were convincingly addressed for 3eSK, this strongly indicates that our responses resolve the major overlapping concerns from GRDX as well.

---

> ### Author Response · Authors · 2025-11-30
>
> # Major Concerns and How We Addressed Them
>
> ---
>
> ### (A) Does using only one example per class introduce bias, limit diversity? Is TrainRef Phase II necessary?
>
>
> **How we addressed it:**
> - Low sensitivity to reference selection: 30 seeds on CIFAR-100 (50% noise) yield 81.92 ± 0.25%.
> - Influence augmentation improves diversity: cosine similarity 0.55 vs 0.67 for nearest-neighbor search.
> - Removing Phase II drops performance by −11.7%, showing it is essential.
>
> **Corresponding questions:**
> 3eSK-Q1, 6MHt-Q2, 6MHt-Q3, GRDX-Q2, GRDX-Q3.
>
> ---
>
> ### (B) Does the computational overhead of TrainRef also increase significantly as the dataset size increases? Is influence computation expensive?
>
> **How we addressed it:**
> - All phases scale linearly with dataset size (a theoretical analysis on time complexity).
> - Influence uses only the linear head, keeping cost small.
> - WebVision 10%→50%→100% subsets scale near-linearly (153→827→1722 min).
>
> **Corresponding questions:**
> R3eSK-Q2, GRDX-Q5.
>
> ---
>
> ### (C) Are human comparisons reliable? Is MIM required?
>
> **How we addressed it:**
> - Human study: 5 experts, 500 judgments, κ ≈ 0.72; samples chosen where methods strongly disagree.
> - MIM not required: using DINOv2 still yields strong results (e.g., CIFAR-100 92.3% at 20% noise).
>
> **Corresponding questions:**
> R3eSK-Q3, R3eSK-Q4.
>
> ---
>
> # Complete Problem List (Map)
>
> **Reviewer 3eSK**
> - Q1: Reference-set selection and potential bias
> - Q2: Scalability and computational overhead
> - Q3: Reliability of human evaluation
> - Q4: Necessity of MIM; alternative SSL backbones
>
> **Reviewer 6MHt**
> - Q1: Categorical vs distributional noise analysis
> - Q2: Diversity of reference set and threshold setting
> - Q3: Necessity of Phase II
> - Q4: Influence threshold and noise on convergence
>
> **Reviewer GRDX**
> - Q1: Expanded discussion and evaluation of normality pollution
> - Q2: Why one sample per class suffices
> - Q3: Dependency on reference quality and coverage
> - Q4: Fairness of comparisons with/without clean sets
> - Q5: Computational overhead in high-class, high-noise settings
> - Q6: Preventing confirmation bias during clean-pool expansion
>
> ---
>
> Thank you again to the reviewers and to you as the Area Chair for your time and dedication.
> We deeply appreciate your careful consideration of our work.

---

### Meta-Review · Area_Chair_BFDj · 2025-12-26

**Summary:**

Since `Reviewer 6MHt`’s questions are primarily clarification-related, and `Reviewer 3eSK ` has indicated that their concerns have been addressed, the following analysis focuses on the remaining concerns raised by `Reviewer GRDX`:

1. Presentation-related suggestions.
2. Sensitivity analysis on how the quality, coverage, or size of the reference set affects performance.
3. Concerns that some baselines do not use an external clean set, potentially rendering the comparisons unfair
4. Model complexity and computational overhead.
5. How the method prevents confirmation bias or the amplification of early errors.
6. Justification for why using “one sample per class” in the reference set is sufficient to avoid normality pollution.

**Reviewer Concerns:**

Based on the rebuttal, the following concerns appear to be addressed:

- The first concern is addressed.
- The second concern is mostly addressed through an analysis of the threshold of the influence score.
- The fourth concern regarding computational complexity appears to be addressed, with analysis and empirical justification provided using results on the WebVision dataset.
- The fifth concern is discussed in the rebuttal.

The following concerns are only partially addressed:

1. The third concern is discussed; however, the comparison may be fair only for the L2B baseline, as the other baselines do not use an external clean label set. The authors should clearly indicate in the experimental tables which methods, including their own, rely on additional clean labels.

2. The final concern is partially addressed. While the authors explain how their method works and provide results on CIFAR-100 with 50% symmetric noise, the AC considers this response only partially sufficient. Given this being a strong claim, further empirical validation across additional noise types and more realistic noise settings, such as those studied in [1], would be beneficial. Some of the remaining claims would benefit from further empirical validation beyond the scope of the rebuttal.

[1] Beyond Synthetic Noise: Deep Learning on Controlled Noisy Labels

**Reviewer Scores:**

- `Reviewer 3eSK` (initial rating: 6) indicated that their concerns have been satisfactorily addressed and increased their score to 8 (accept).
- `Reviewer 6MHt` (initial rating: 8) is likely to maintain their positive score.
- `Reviewer GRDX` (initial rating: 4) is likely to increase their score (e.g., to 6), as their concerns are partially addressed.

So the extrapolated final reviews are 8, 8, 6.

---

### Decision · Program_Chairs · 2026-01-26

Accept (Poster)